# A corticostriatal deficit promotes temporal distortion of automatic action in ageing

Miriam Matamales[1,2], Zala Skrbis[1], Matthew R Bailey[3], Peter D Balsam[4], Bernard W Balleine[2], Jürgen Götz[1], Jesus Bertran-Gonzalez[1,2]*

[1]Clem Jones Centre for Ageing Dementia Research, Queensland Brain Institute, The University of Queensland, Brisbane, Australia; [2]Decision Neuroscience Laboratory, School of Psychology, University of New South Wales, Sydney, Australia; [3]Psychology Department, Columbia University, Broadway, United States; [4]Psychology Department, Barnard College, Columbia University, Broadway, United States

**Abstract** The acquisition of motor skills involves implementing action sequences that increase task efficiency while reducing cognitive loads. This learning capacity depends on specific cortico-basal ganglia circuits that are affected by normal ageing. Here, combining a series of novel behavioural tasks with extensive neuronal mapping and targeted cell manipulations in mice, we explored how ageing of cortico-basal ganglia networks alters the microstructure of action throughout sequence learning. We found that, after extended training, aged mice produced shorter actions and displayed squeezed automatic behaviours characterised by ultrafast oligomeric action chunks that correlated with deficient reorganisation of corticostriatal activity. Chemogenetic disruption of a striatal subcircuit in young mice reproduced age-related within-sequence features, and the introduction of an action-related feedback cue temporarily restored normal sequence structure in aged mice. Our results reveal static properties of aged cortico-basal ganglia networks that introduce temporal limits to action automaticity, something that can compromise procedural learning in ageing.

DOI: https://doi.org/10.7554/eLife.29908.001

*For correspondence:
j.bertran@unsw.edu.au

**Competing interests:** The authors declare that no competing interests exist.

## Introduction

Learning of new skills permits optimal interactions with the environment while reducing cognitive costs, a fundamental adaptation contributing to behavioural autonomy and automaticity in many species. Motor skills are generally implemented through sequence learning, which allows elementary action units to be integrated into behavioural streams (*Lashley, 1951*), reducing latency and increasing speed of action (*Sternberg et al., 1978*). Similar to memory, movement sequences are thought to be organised into 'motor chunks', a cognitive-motor strategy proposed to depend on cortico-basal ganglia circuitry (*Graybiel, 1998*; *Wymbs et al., 2012*) that allows expression of whole behavioural programs as a single response (*Abrahamse et al., 2013*).

Although initial learning capabilities are often preserved (*Voelcker-Rehage, 2008*; *Matamales et al., 2016a*), motor skill capacity is reduced in older adults, especially in tasks involving fine motor control. Indeed, some studies report that aged humans have a near zero capacity to form action chunks in newly acquired sequences (*Shea et al., 2006*; *Verwey, 2010*), whereas others find evidence of shortened motor chunks (*Bo et al., 2009*). Sequence learning appears, therefore, to be impaired in ageing, although the acquisition of new skills may still be possible, perhaps through the recruitment of compensatory strategies (*Cabeza et al., 2002*; *Park and Reuter-Lorenz, 2009*).

Nevertheless, despite increasing knowledge of the neural bases of action automatisation (*Jin and Costa, 2015*), the precise neuronal mechanism by which advanced ageing affects the acquisition of skills remains unknown.

To address this issue, we first designed behavioural tasks to determine which elements of action sequence learning were impaired by ageing. We then identified neuronal activation changes in the corticostriatal network that correlated with age-related learning defects, and reproduced some of these behavioural features through circuit-specific manipulations in young transgenic mice. Finally, we explored approaches to restore the ability of aged mice to perform action sequences through the introduction of an action-related feedback cue. Our findings suggest that the ageing brain may engage particular neuronal strategies to automatise action during skill learning, and that environmental support in the form of action-related feedback can be used to correct this behaviour.

## Results

### Aged mice display shorter patterns of action

We first compared the ability of young (2 months old) and aged (20–22 months old) mice to develop action sequences. We focused on homogeneous sequences involving the concatenation of single responses because this procedure is particularly susceptible to chunking (*Garcia-Colera and Semjen, 1987*; *Jin and Costa, 2015*). After mild food restriction, we trained animals using an instrumental procedure in which the number of lever presses (LP) required to obtain food outcomes increased as training progressed on a random ratio schedule of reinforcement (from constant reinforcement, CRF, to random ratio 20, RR20; *Figure 1A*). The formation of action sequences was promoted by introducing a sequence trigger (ST), in which caching of 5 or 7 contiguous LP was required to access the current RR program (see *Figure 1—figure supplement 1A*). We used a novel approach to monitor the formation of sequences by parsing behavioural data into three elements: initiation (first LP, or LP occurring after a magazine check), execution (LP occurring after a LP) and termination (magazine check occurring after LP) (*Figure 1B*). Throughout training, both young and aged mice significantly escalated their overall performance (LP rate) as the ratio requirement to obtain an outcome increased (mixed ANOVA using factors of training and age $F_{(3.5,48.9)}$ = 116.852, p<0.001). A non-significant training x age interaction ($F_{(3.5,48.9)}$ = 1.153, p=0.341) suggested that overall instrumental performance was equal in young and aged mice (*Figure 1—figure supplement 1B*). However, half way through this training, aged mice produced a larger number of action sequences per minute, whereas young mice stabilised, or diminished, their sequence rates (*Figure 1C*). This was supported by a significant effect of training ($F_{(16,224)}$= 28.885, p<0.001) and a significant training x age interaction ($F_{(16,224)}$ = 3.716, p<0.001). Further, a comparison of action sequences between days 7 and 17 of training revealed that both young and aged mice were able to extend the number of execution elements in their sequences throughout training ($F_{(1,14)}$ = 158.282, p<0.001) (*Figure 1D*). This increase was of a different magnitude (as suggested by a significant day x age interaction ($F_{(1,14)}$ = 40.03, p<0.001)), indicating that aged mice extended their sequences to a lesser extent than young mice. Separate analysis confirmed, however, that, although at a different level, both young ($F_{(1,7)}$ = 108.562, p<0.001) and aged ($F_{(1,7)}$ = 106.664, p<0.001) groups showed significant increases in sequence length (*Figure 1D*). In contrast, analysis of sequence duration showed a significant day x age interaction ($F_{(1,14)}$ = 8.368, p<0.05) with significant increases of sequence duration in young ($F_{(1,7)}$ = 18.376, p<0.01) but not aged ($F_{(1,7)}$ = 1.745, p=0.228) mice (*Figure 1E*). Importantly, aged mice showed evidence of learning about the ST applied from day 10, as they displayed a marked increase in the number of sequences that broke through the ST and therefore frequently accessed the RR schedule (*Figure 1—figure supplement 1C*). In contrast, young mice kept these numbers low, suggesting that they comfortably incorporated the ST requirement ($F_{(3.065,42.908)}$ = 7.757, p<0.001; training x age interaction $F_{(3.065,42.908)}$ = 6.027, p<0.01) (*Figure 1—figure supplement 1C*). Event-time plots recorded in both groups on day seven revealed no sequence structure in their instrumental performance, despite all rewards being obtained by all of the mice (*Figure 1—figure supplement 1D*). In contrast, analysis on day 17 revealed that both young and aged mice developed clearly defined action sequences over time (with identifiable initiation, execution and termination elements), although these sequences were more frequent and much shorter in the aged group (*Figure 1F*).

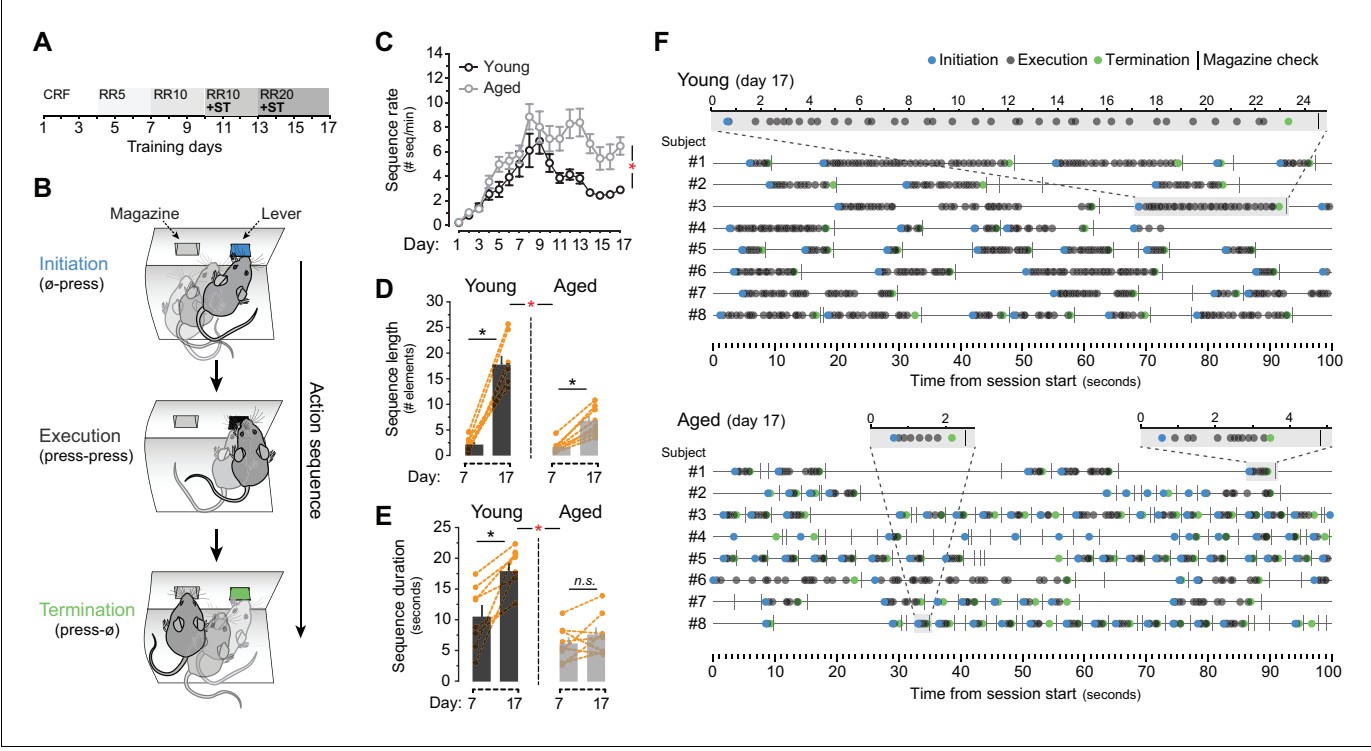

**Figure 1.** Evidence of sequence learning in aged mice. [*Figure 1—source data 1*] (A) Experimental design of instrumental conditioning under an increasing random ratio (RR) schedule of reinforcement, from constant reinforcement (CRF) to RR20. Sequence triggering (ST) programs were introduced on day 10. (B) Initiation, execution and termination elements were identified in each sequence according to lever press (LP) and magazine entry interspersing. (C) Number of action sequences per minute (sequence rate) displayed throughout instrumental learning [StatsReport1]. (D, E) Average length (D) and duration (E) of LP sequences produced by each young and aged mouse on days 7 and 17 of training. Orange traces represent individual mice. Asterisks denote significant day x age interaction (red) and simple effects (black, see text). N.S., not significant [StatsReport2] [StatsReport3]. (F) Event-time diagrams showing linearly organised LP sequences produced by all young and aged mice on training day 17. Shaded segments (gray) are amplified LP sequences (seconds). See *Figure 1—figure supplement 1D* for pre-sequence diagrams.
DOI: https://doi.org/10.7554/eLife.29908.002

The following source data and figure supplements are available for figure 1:

**Source data 1.** Source data for *Figures 1* and *3*.
DOI: https://doi.org/10.7554/eLife.29908.005
**Source data 2.** Source data for *Figure 1*.
DOI: https://doi.org/10.7554/eLife.29908.006
**Source data 3.** Source data for *Figure 1*.
DOI: https://doi.org/10.7554/eLife.29908.007
**Figure supplement 1.** Application of a sequence trigger (ST) to promote sequence learning.
DOI: https://doi.org/10.7554/eLife.29908.003
**Figure supplement 2.** Pellet drop detection in operant conditioning chambers.
DOI: https://doi.org/10.7554/eLife.29908.004

We next ruled out that this effect was due to the inability of aged mice to hear when pellets were delivered (*Figure 1—figure supplement 2*). First, we confirmed that the delivery of pellets in the operant conditioning boxes generated a clear vibration signal, which was more salient than the sound of the dispenser engagement and/or pellet dropping (*Figure 1—figure supplement 2A*). We then exposed a group of young and aged mice to six days of magazine training, where 20 pellets were randomly delivered in ~30 min sessions (*Figure 1—figure supplement 2B–C*). We found that all mice reduced their reaction time between pellet delivery and first magazine check (first check intervals) as training progressed (mixed ANOVA with factors training and age $F_{(5,70)}$ = 25.822, p<0.001), indicating that they learned to detect when pellets were delivered. A non-significant training x age interaction ($F_{(5,70)}$ = 0.360, p=0.874) confirmed that this reduction was similar in both

groups (*Figure 1—figure supplement 2B and D*). This similar performance was not due to a higher overall number of magazine checks in the aged group, since both young and aged mice showed a similar increase of magazine approach responses throughout this training ($F_{(5,70)}$ = 25.822, p<0.001; training x age interaction $F_{(5,70)}$ = 0.360, p=0.874) (*Figure 1—figure supplement 2C*). Overall, our results showed that aged mice produced a higher number of action sequences, and in contrast to previous studies in humans (*Shea et al., 2006*; *Verwey, 2010*), these sequences could be extended (in terms of action number) but not temporally sustained.

We thus explored whether sustained actions in aged mice were subject to temporal constraints, something that could explain the formation of shorter sequences of action. To address this, we used a Lever Hold (LH) instrumental procedure (*Figure 2*), a variant of the variable interval hold task (*Bailey et al., 2015*) in which the delivery of the reward depended on pseudorandom increments of LH time during training (*Figure 2A and B*). In this task, both young and aged groups displayed parallel escalations of LH times up to ~0.5 s. However, young mice increased their LH times throughout subsequent stages of training, whereas aged mice maintained their maximum performance around 0.5 s holds (*Figure 2C*). Statistical analysis supported this observation: mixed ANOVA (factors: training and age) showed that the time of lever hold significantly varied as instrumental training progressed in both age groups ($F_{(16,176)}$ = 73.471, p<0.001), although there was a significant training x age interaction, indicating that the age of the animals significantly affected their performance ($F_{(16,176)}$ = 16.796, p<0.001). Importantly, simple effects analyses confirmed that despite these differences, both groups independently elevated their performance throughout training (Young: $F_{(16,112)}$ =

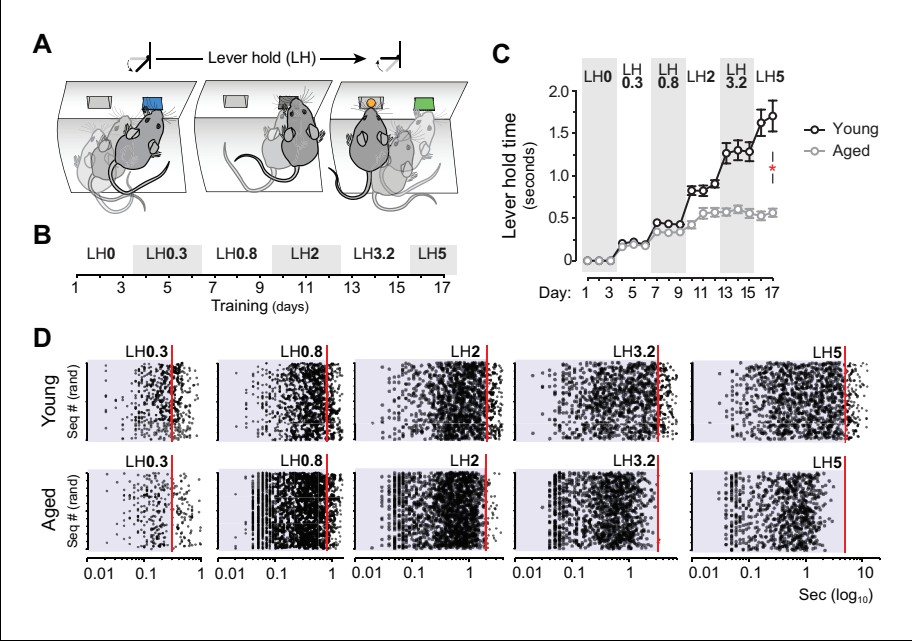

**Figure 2.** Aged mice are unable to temporally sustain their actions. [*Figure 2—source data 1*] [StatsReport4] (**A**) Mice were trained to press and hold the lever down for increasing periods to obtain rewards. (**B**) Experimental design of instrumental conditioning under an increasing pseudorandom lever hold (LH) schedule of reinforcement. Average LH times are indicated in seconds. (**C**) Acquisition of LH instrumental behaviour across training. Data are mean ±SEM (7 and 8 mice per group). Asterisk denotes significant training x age interaction. (**D**) Representation of each individual LH performed by all young and all aged mice on the last day of LH0.3, LH0.8, LH2, LH3.2 and LH5 training. LH times are presented in a Log10 scale (x axis), and sequences are distributed randomly in the y axis.
DOI: https://doi.org/10.7554/eLife.29908.008

The following source data and figure supplement are available for figure 2:

**Source data 1.** Source data for *Figure 2*.
DOI: https://doi.org/10.7554/eLife.29908.010
**Figure supplement 1.** Aged mice produce shorter lever holds.
DOI: https://doi.org/10.7554/eLife.29908.009

69.387, p<0.001; Aged: $F_{(16,64)}$ = 51.33, p<0.001). Event scatter plots at different LH requirements revealed that, although both groups displayed a large number of very short presses, only young mice managed to produce sustained presses that approached, or exceeded, each LH requirement (*Figure 2D*). Event-time diagrams showed how all aged mice displayed much shorter holds than younger controls from the start of the session (*Figure 2—figure supplement 1*).

As such, these results provide evidence of action sequence formation in aged mice while also suggesting that these sequences could be influenced by temporal constraints, in line with previous results in humans (*Bo et al., 2009*).

## Sequence learning leads to rapid response bursts and aberrant chunking in aged mice

We next investigated the forms of automaticity generated in both age groups during LP-based sequence learning. First, we analysed in our sequence timestamp dataset whether aged mice showed higher overall sequence speed. We found that average speed on the last day of training was higher in aged mice, although the effect fell below significance ($t_{(14)}$ = −2.104, p=0.054) (*Figure 3A*). However, subsequent analysis of the serial response times based on inter-press-intervals (IPIs) recorded on the same day revealed that aged mice produced a much higher proportion of very short IPIs (*Figure 3B*, *Figure 3—figure supplement 1A*), suggesting that they were reaching a higher sequence speed than young mice. Due to a resolution limit in the timestamp data (*Figure 3—figure supplement 1B*), we sought to establish the true maximum sequence speed by means of acoustic data, where the fastest bursts of LP in each mouse were identified by extracting the acoustic fingerprint of lever depression (*Figure 3C*, see Materials and methods). We found a strikingly high speed of responding in aged mice, in some instances displaying spates of LP with over 13 presses per second, much faster than the fastest bursts recorded in young mice (*Figure 3C*, *Video 1* and *Video 2*). These speeds are similar to fully automatised scratch reflex movements in mice (*Inagaki et al., 2003*). Quantification analysis of fast sequences confirmed that average speed within fast bursts (in number of presses/sec) was significantly higher in aged than young mice ($t_{(7.417)}$ = −4.201, p<0.01, *Figure 3D*), whereas within-burst IPIs were significantly shorter ($t_{(14)}$ = 2.724, p<0.05, *Figure 3E*). Interestingly, despite their speed of responding, aged mice were less efficient at earning rewards ($t_{(14)}$ = 3.101, p<0.01), indicating that although their action bursts often broke through the sequence trigger (see *Figure 1—figure supplement 1C*), they rarely completed the RR requirements (*Figure 3F*).

We next sought to analyse whether different chunking patterns emerged in young and aged mice during sequence learning by studying the temporal relationships between consecutive action elements throughout training (*Figure 3G–K*). We categorised the action repertoire into two classes of interval: the within-sequence interval (IPIs that reside within each sequence) and the sequence boundary interval (a combination of press-check and check-press intervals that lay in-between sequences) (see methods). The analysis of the distribution of these intervals at different stages of training provided a visual readout of the chunking that emerged during sequence development (*Figure 3G*). In both young and aged mice, within-sequence IPIs (*Figure 3G*, blue) clustered around a similar interval space (from ~0.1 to ~1 s intervals) by day 13, suggesting that both groups developed chunking of within-sequence (i.e. execution) elements. In support of this, we found that both young and aged mice similarly reduced the within-sequence IPIs throughout training (*Figure 3H*), as reflected by a significant effect of training (two-way mixed ANOVA with factors training and age: $F_{(1.955,27.377)}$ = 18.606, p<0.001) but no training x age interaction ($F_{(1.955,27.377)}$ = 0.562, p=0.573). Similarly, the variability of these IPIs was equally reduced in both groups (*Figure 3H*, inset), again showing a significant effect of training ($F_{(2.352,32.904)}$ = 6.545, p=0.003) with no training x age interaction ($F_{(2.352,32.904)}$ = 1.785, p=0.179). On the other hand, the development of sequence boundary intervals throughout training (*Figure 3G*, orange) was very different in young and aged mice. Whereas young mice displayed longer boundary intervals that were distant from the chunking space, aged mice tended to accumulate them in a narrow band near the chunking space (*Figure 3G*). Importantly, this band appeared earlier in training (day 10) in all aged animals, before within-sequence intervals chunked (*Figure 3G*, *Figure 3—figure supplement 2A and B*). This was evidenced in the quantitative analysis, where the young group showed an increase in sequence boundary intervals from days 10 to 17, whereas the aged group kept these intervals consistently low (*Figure 3I*), as indicated by a significant effect of training ($F_{(3.884,54.375)}$ = 8.005, p<0.001) and a

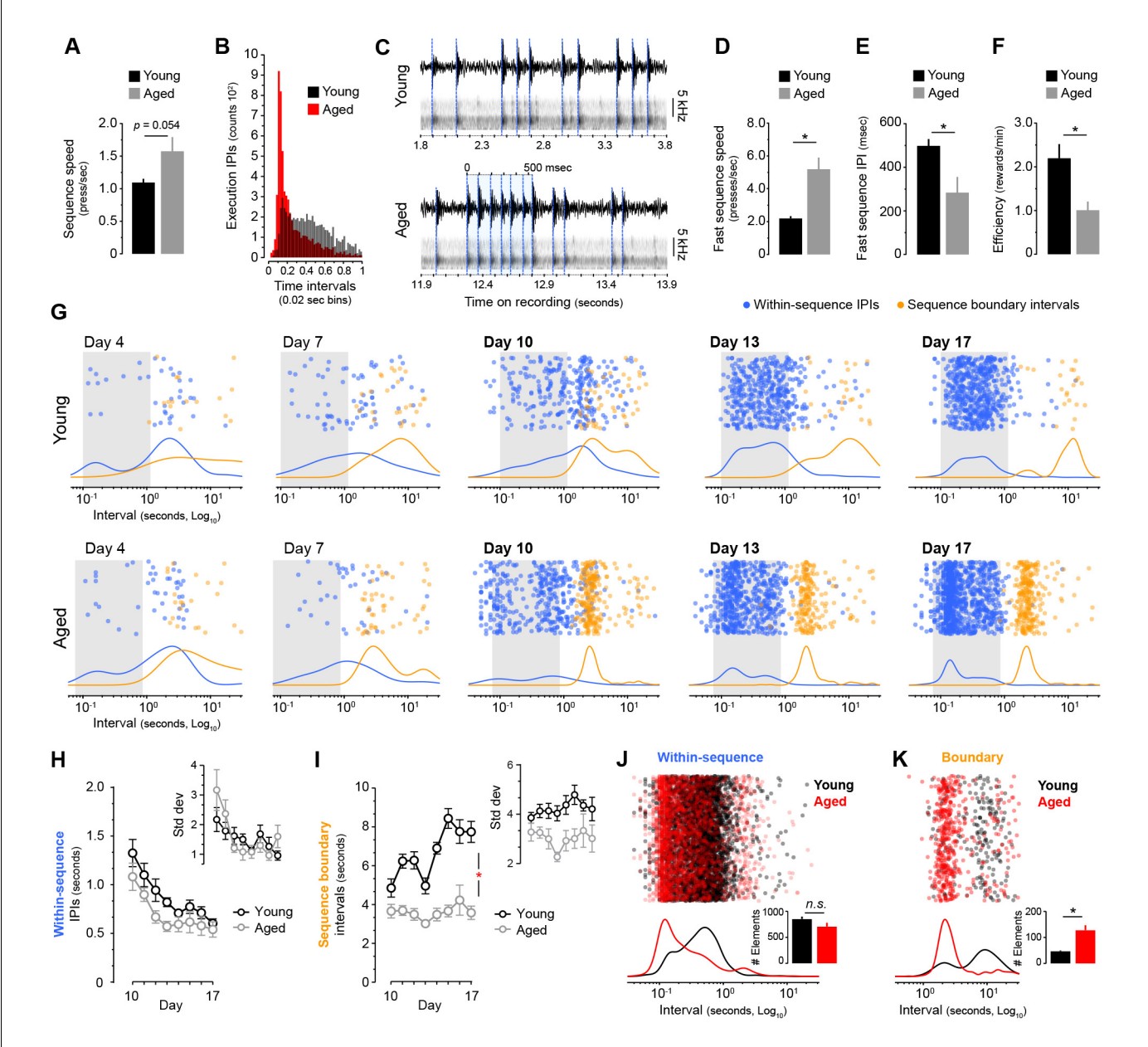

**Figure 3.** Aged mice produce ultrafast sequences and aberrant chunking. [*Figure 1—source data 1*] [Figure 3—source data 3] (**A**) Average sequence speed calculated from timestamp data after 20 min of continuous instrumental performance on training day 18 [StatsReport33]. (**B**) Frequency distribution of the number of inter-press-intervals (IPIs) recorded in all young and aged mice at increasing intervals on training day 18. (**C**) Real IPIs from the fastest sequence recorded in a young and an aged mouse on day 18 (Fast sequence one in *Video 1* and *Video 2*). Real lever press times were identified by aligning the sound waveform (top) and acoustic spectrogram (bottom) generated by lever press activity during the session. Identified presses are marked with a blue dashed line. (**D, E**) Average speed (**B**) and IPIs (**C**) of the five fastest LP sequences displayed by young and aged mice during the 20 min session on day 18 extracted from acoustic data [StatsReport5] [StatsReport6]. (**F**) Efficiency of LP instrumental behaviour (rewards earned per minute) in young and aged mice on day 18 [StatsReport7]. Asterisks (**B–D**) denote significant effects (see text). (**G**) Scatter plots of each within-sequence inter-press-interval (IPIs, blue) and sequence boundary interval (orange) produced by one young and one aged mouse (Young #7 and Aged #5) at different stages of training. Kernel density curves of each interval type are plotted at the bottom. Grey shades delimit the within-sequence chunking space. (**H, I**) Average within-sequence IPIs (**G**) and sequence boundary (**H**) intervals. Insets are corresponding standard deviations. Data are mean ±SEM. Asterisk denotes significant training x age interaction [StatsReport27] [StatsReport28] [StatsReport31] [StatsReport32]. (**J, K**) Proportional scatter plots and kernel density curves showing intervals of the within-sequence (**I**) and sequence boundary (**J**) elements produced by all young and all aged mice during the first 600 s of training day 18. Insets are the total element counts for this period [StatsReport29] [StatsReport30]. Data in A, D-F, J and K are mean +SEM. *p*, p-value; asterisk, significant effect; N.S., not significant.

DOI: https://doi.org/10.7554/eLife.29908.011

*Figure 3 continued on next page*

*Figure 3 continued*

The following source data and figure supplements are available for figure 3:

**Source data 1.** Source data for *Figure 3*.
DOI: https://doi.org/10.7554/eLife.29908.014
**Figure supplement 1.** Ultra-fast IPIs in aged mice.
DOI: https://doi.org/10.7554/eLife.29908.012
**Figure supplement 2.** Sequence boundary intervals in aged mice are regularised early in training.
DOI: https://doi.org/10.7554/eLife.29908.013

significant training x age interaction ($F_{(3.884,54.375)}$ = 4.729, p=0.003). Although no drop in variability for this interval was found from day 10 to 17 ($F_{(7,98)}$ = 1.428, p=0.203), aged mice displayed lower variability than young mice in this period ($F_{(1,14)}$ = 8.855, p=0.01) (*Figure 3I*, inset), suggesting that the temporal relationships of boundary elements in aged mice had already been established in day 10 (see *Figure 3—figure supplement 2*). In order to obtain proportional measures of chunking in both groups, we compared the within-sequence and sequence boundary intervals produced by all young and all aged mice during the first 600 s of continuous action sequence behaviour (*Figure 3J–K*). Again, we found in aged mice a larger proportion of very short within-sequence IPIs (*Figure 3J*), although the overall number of within-sequence presses was equal across groups ($t_{(14)}$ = 1.692, p=0.113) (*Figure 3J*, inset). The sequence boundary intervals produced during this period were also shorter in aged mice (*Figure 3K*), and the larger number of boundary elements in this group ($t_{(7.341)}$ = −4.248, p=0.003) (*Figure 3K*, inset) reflected the higher number of action sequences produced by aged mice per minute (see *Figure 1C*). Therefore, we found clearly different action automatisation processes shaping within-sequence and sequence boundary elements in young and aged mice. While young mice appeared to exclusively chunk the elements within the sequences, aged mice displayed ultrafast chunking of within-sequence elements, and showed evidence of early chunking in sequence boundary elements. Together, these results suggest that early chunking of the extra sequence elements in aged mice could cleave behaviour into ultrafast microchunks, revealing previously unrecognised deficits in automatic action in ageing.

## Sequence features correlate with defective development of corticostriatal activity in aged mice

The development of action sequences and stimulus-response habits is attributed to mediolateral shifts of activity in cortico-basal ganglia circuits (*Balleine, 2005*; *Yin et al., 2009*; *Balleine and O'Doherty, 2010*), likely following a medial-to-lateral disposition of functionally relevant corticostriatal projections in rodents (*Hunnicutt et al., 2016*; *Voorn et al., 2004*). Here, we studied the state of activation of the corticostriatal network at different rostrocaudal levels in young and aged mice expressing uninterrupted action sequences for 20 min (*Figure 4*). We identified over 60,000 activated neurons across cortical and striatal regions in sixteen mice using a confocal imaging-computational approach based on the cytonuclear detection and mapping of phosphorylated MAPK ERK1/2 (p-MAPK) (*Figure 4A and B*), a marker that is widely used to study neuronal activity throughout the brain (*Valjent et al., 2004*). We first contrasted the extent of activation detected in cortical and striatal regions in young and aged mice (*Figure 4C*). Two-way ANOVA (factors: age; region) revealed no overall effect of age ($F_{(1, 111)}$=1.868, p=0.174), although there was a significant age x region interaction ($F_{(7,111)}$ = 2.136, p<0.05). Simple effect analyses (factor: age) showed that this interaction was primarily driven by differences in

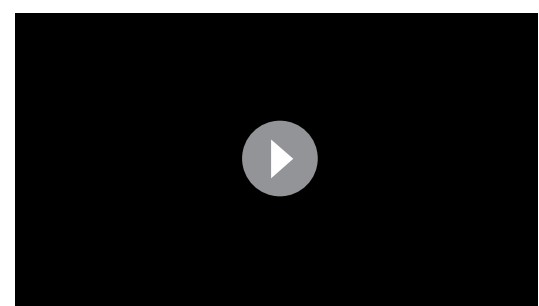

**Video 1.** Representative performance of a young mouse during the last day of sequence training. Two fast action sequences are captured (indicated in red font, top left corner). Fast action sequence one is represented in *Figure 3A*. Subject Young#5; active lever: right; training day 18.
DOI: https://doi.org/10.7554/eLife.29908.015

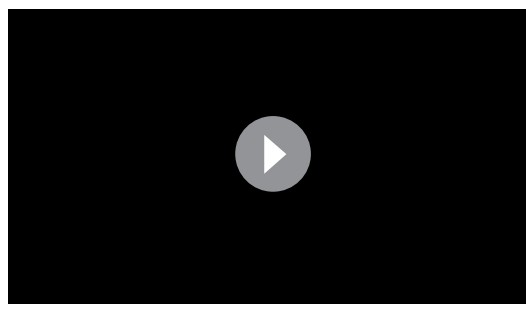

**Video 2.** Representative performance of an aged mouse during the last day of sequence training. Two fast action sequences are captured (indicated in red font, top right corner). Fast action sequence one is represented in *Figure 3A*. Subject Aged#4; active lever: left; training day 18.
DOI: https://doi.org/10.7554/eLife.29908.016

the anterior cingulate/motor area 2 (aCg/M2; $F_{(1,14)} = 4.912$, $p<0.05$) and the frontoparietal regions of the cortex (FPx: $F_{(1,14)} = 15.364$, $p<0.01$), as well as in their target territories of the dorsolateral striatum (DLS: $F_{(1,13)} = 4.753$, $p<0.05$) (*Ebrahimi et al., 1992*; *McGeorge and Faull, 1989*). We next used this extensive dataset to identify the corticostriatal regions that were primarily related to action sequence execution. We conducted a correlation study by contrasting the corticostriatal activity profile based on p-MAPK activation in different cortical and striatal areas in each mouse with the different features of sequence performance. We found that sequence length, duration and speed of action displayed on the last day of training could only be predicted by activation in DLS and FPx, whereas activation in all other areas did not significantly correlate with any feature of sequence execution (*Figure 4D*). We then investigated the relative functional engagement of different territories across the corticostriatal network in each age group by performing a regional cross-correlation analysis in which the mean level of p-MAPK activation in each region was correlated with all other regions across animals (*Figure 4E*). Young mice showed very low levels of correlation overall, with no significant relationships between cortical and striatal territories throughout the rostrocaudal extent of the network, perhaps due to automaticity-related reductions in neuronal variability and/or processing loads (*Santos et al., 2015*; *Dayan and Daw, 2008*) (*Figure 4E*, left panel). In contrast, aged mice showed widespread correlations that were significant in different regions of the corticostriatal network, especially the rostromedial, but not caudolateral, territories (*Figure 4E*, right panel). In order to obtain further regional information on this effect, we compared the spatial distribution of activated neurons in the corticostriatal network by mapping the position of individual neurons in each territory of sections from young and aged mice (*Matamales et al., 2016b*) (*Figure 4F*). Our data in aged mice revealed large regions within the DLS and deep layers of the FPx that were devoid of neuronal activation (*Figure 4F*). Overall, these results suggested an age-related deficit in the posterior corticostriatal network during action sequence execution that principally involved functional impairments in dorsolateral striatum and frontoparietal cortex, the two regions primarily implicated in motor chunking (*Wymbs et al., 2012*).

## Chemogenetic inhibition of direct pathway projection neurons in dorsolateral striatum increases speed of action sequences

Striatal output circuits are organised into direct and indirect pathway neurons (dSPNs and iSPNs, respectively), two parallel projection systems that orchestrates functions in the basal ganglia (*Gerfen and Surmeier, 2011*). Recent studies have suggested that action duration could be influenced by dSPNs in the DLS based on their sustained firing during the time of sequence execution (*Jin et al., 2014*), and the prolongation of an ongoing action by optogenetic stimulation (*Tecuapetla et al., 2016*). Capitalising on our p-MAPK mapping results, and based on findings that this intracellular activity marker is more sensitive in dSPNs than iSPNs (*Bertran-Gonzalez et al., 2009*), we hypothesised that age-related hypoactivation of dSPNs in the DLS could be contributing, at least in part, to deficits in action sequence learning observed in aged mice. With the aim of reproducing the molecular deficit observed in DLS striatal neurons of aged mice throughout sequence learning, we used M4-based chemogenetics ($G\alpha_i$) to prevent intracellular signalling in dSPNs of the DLS in young transgenic mice at the time of sequence implementation (i.e. day 10 onwards). *Drd1a*-Cre:*Drd2*-eGFP double transgenic mice were given a unilateral intra-DLS infusion of an AAV vector containing the $G\alpha_i$-hM4Di DREADD with Cre specificity coupled to the reporter mCherry, a procedure that allowed manipulation and visualisation of dSPNs and iSPNs, respectively (*Figure 5—figure supplement 1*). In order to test the validity of this system, we combined the injection of CNO (3 mg/kg, i.p.) with a delayed injection of GBR12783 (15 mg/kg, i.p.), a specific dopamine reuptake blocker

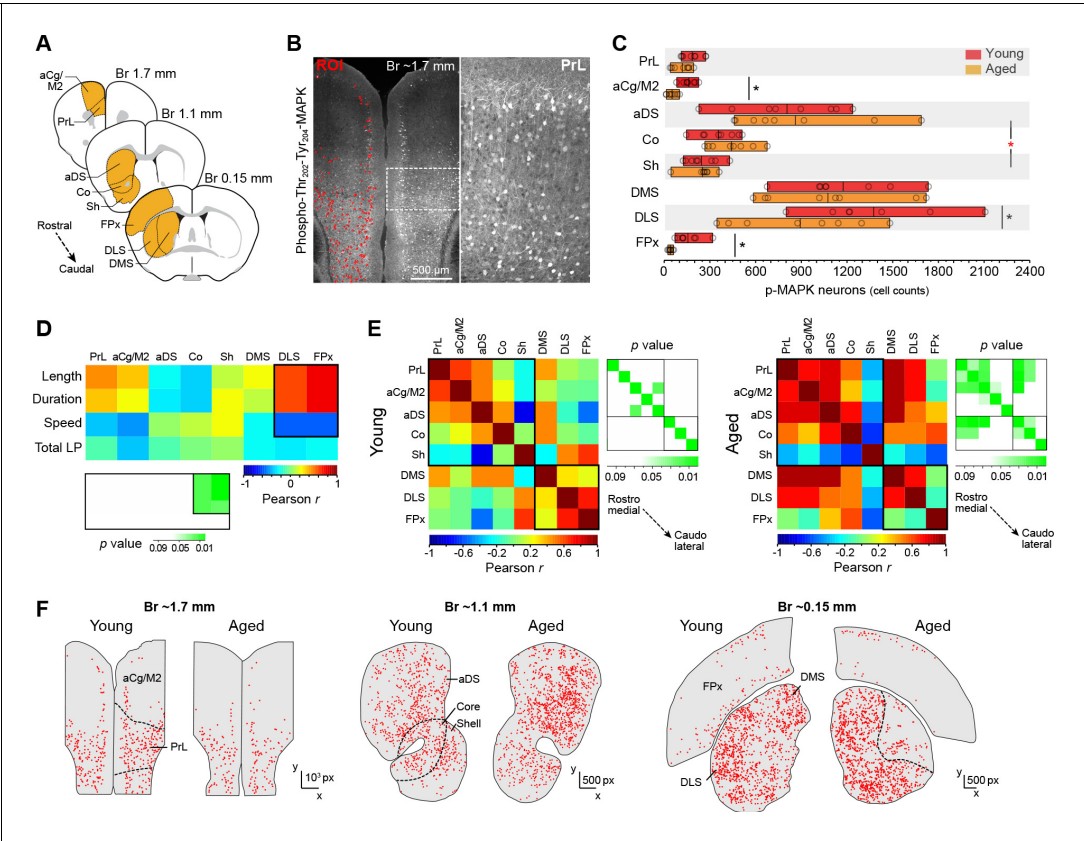

**Figure 4.** Action sequence features correlate with defective activation of caudolateral corticostriatal networks in aged mice. [*Figure 4—source data 1*] (A) Extensive analysis of neuronal activation in different rostrocaudal cortical (prelimbic [PrL], anterior cingulate/motor area 2 [aCg/M2], frontoparietal [FPx]) and striatal (anterodorsal [aDS], core, shell, dorsomedial [DMS], dorsolateral [DLS]) areas after 20 min of uninterrupted instrumental performance (day 18). (B) Position and intensity of each activated neuron (p-Thr$_{202}$-Tyr$_{204}$-MAPK-immunoreactive [p-MAPK]) were detected through semi-automated quantitative fluorescence (~60,000 neurons analysed). (C) Horizontal floating bar chart displaying the quantification of p-MAPK counts in different cortical and striatal areas of young and aged mice. Minimum, maximum and mean of the data are represented (vertical lines). Circles indicate individual counts for each animal (eight mice per group). Asterisks denote significant age x region interaction (red) and simple effects (black, see text) [StatsReport9]. (D) Activity – performance correlation study comparing the counts of p-MAPK detected in the different cortical and striatal areas (arranged from rostro-medial to caudolateral) and the action sequence features (length, duration and speed) as well as overall instrumental performance (total number of lever presses) displayed by each young and aged mouse during the last day of training. The corresponding p value matrix (bottom inset) indicates the significance of each correlation. Pearson r (multicolor scale) and p (green scale) value diagrams are shown for each correlation [StatsReport10]. (E) Cross-correlation analysis of the p-MAPK activity recorded in the different areas (arranged from rostromedial to caudolateral) in all young (left) and aged (right) mice. Multi-colored diagrams show the extent of correlation between each pair of regions (Pearson r values), and green diagrams (right) indicate their significance at each position (p values) [StatsReport11]. Correlations in D and E involve within-subjects measures. (F) Digitised reconstructions of the distribution of activated neurons in corticostriatal regions at rostral (left), medial (middle) and caudal (right) levels in one young and one aged mouse. Each dot represents the position of one neuron expressing p-MAPK.

DOI: https://doi.org/10.7554/eLife.29908.017

The following source data is available for figure 4:

**Source data 1.** Source data for *Figure 4*.
DOI: https://doi.org/10.7554/eLife.29908.018

known to induce p-MAPK in dSPNs (*Valjent et al., 2010*) (*Figure 5—figure supplement 1B*). We found that hM4Di-mCherry expression was contained, within the DLS, to eGFP-negative neurons, confirming specific infection of dSPNs in the DLS (*Figure 5—figure supplement 1C,D*). Importantly, dSPN-hM4Di expression and subsequent activation by CNO prevented GBR-induced MAPK phosphorylation when compared to the non-injected hemisphere of the same animals, ensuring that this system prevents intracellular activation (*Figure 5—Figure 1D*). Quantification of p-MAPK +neurons confirmed this effect (*Figure 5*—figure supplement E): paired comparisons revealed a pronounced

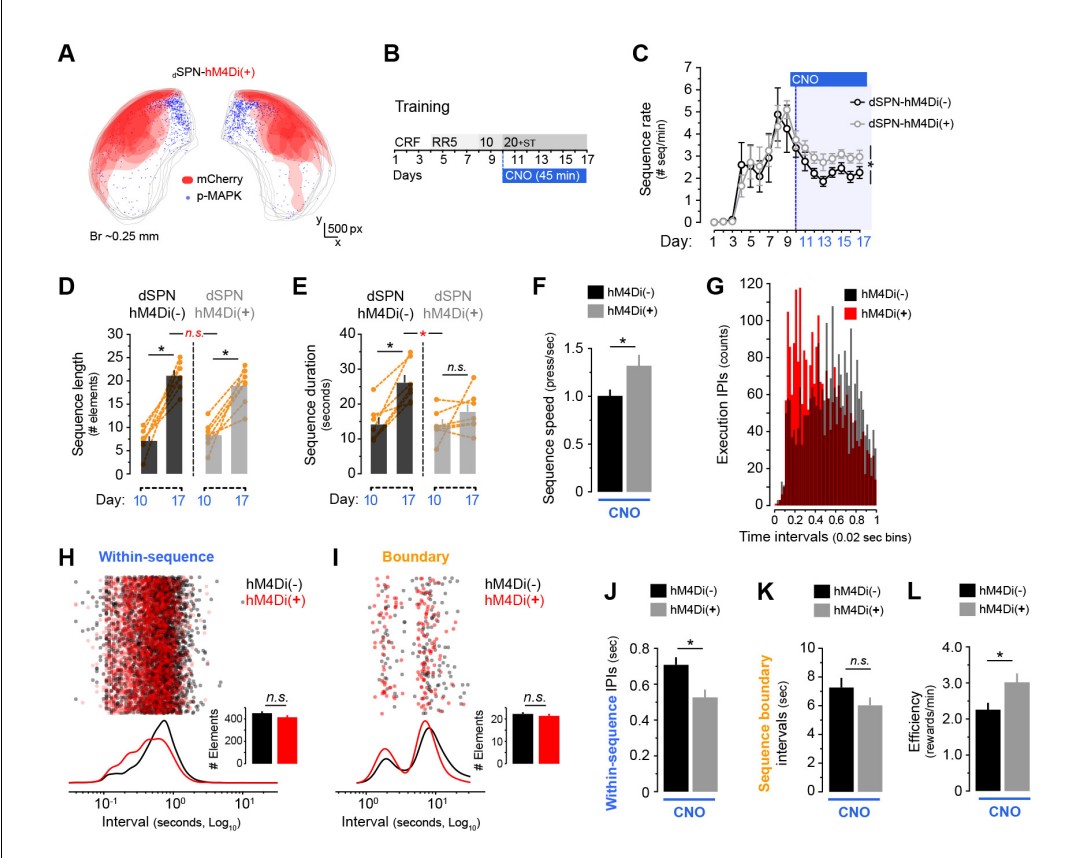

**Figure 5.** Chemogenetic inhibition of dorsolateral direct pathway neurons increases within-sequence speed. [*Figure 5—source data 1*] A cohort of 7 *Drd1a*-Cre[(−)]:*Drd2*-eGFP and 7 *Drd1a*-Cre[(+)]:*Drd2*-eGFP young mice (dSPN-hM4Di[(−)] and dSPN-hM4Di[(+)]) were bilaterally injected with AAV2-hSyn-DIO-hM4Di-mCherry in the DLS. (A) Diagram shows the overlapping spread of viral infection in all dSPN-hM4Di[(+)] mice throughout the DLS as visualised by mCherry-expressing territories. Blue dots represent mapped p-MAPK-positive neurons in the same sections. (B) Upon recovery, all mice were submitted to LP sequence training with daily CNO injections from day 10 (3 mg/kg, 45 min before each session). (C) Number of action sequences per minute (sequence rate) displayed throughout instrumental learning. Data are mean ±SEM. Asterisk denotes significant between subjects effect across CNO period (see text) [StatsReport37]. (D, E) Average length (D) and duration (E) of LP sequences produced by each mouse on days 10 and 17 of training. Orange traces represent individual mice. Asterisks denote significant day x group interaction (red) and simple effects (black, see text) [StatsReport13] [StatsReport14]. (F) Average speed of the LP sequences displayed by dSPN-hM4Di[(−)] and dSPN-hM4Di[(+)] on day 18 (timestamp data) [StatsReport15]. (G) Frequency distribution of the number of inter-press-intervals (IPIs) recorded in all dSPN-hM4Di[(−)] and dSPN-hM4Di[(+)] mice at increasing intervals on training day 18. (H, I) Scatter plots and kernel density curves showing intervals of the within-sequence (H) and sequence boundary (I) elements produced by all dSPN-hM4Di[(−)] and dSPN-hM4Di[(+)] on training day 18. Insets are the total element counts for this period [StatsReport35]. (I, J) Average within-sequence IPIs (I) and sequence boundary (J) intervals on day 18. Data are mean +SEM [StatsReport16] [StatsReport36]. (K) Efficiency of LP instrumental behaviour (rewards earned per minute) in by dSPN-hM4Di[(−)] and dSPN-hM4Di[(+)] mice on day 18 [StatsReport17]. Asterisks denote significant effects (see text). N.S., not significant.

DOI: https://doi.org/10.7554/eLife.29908.019

The following source data and figure supplements are available for figure 5:

**Source data 1.** Source data for *Figure 5*.
DOI: https://doi.org/10.7554/eLife.29908.022
**Source data 2.** Source data for *Figure 5*.
DOI: https://doi.org/10.7554/eLife.29908.023
**Figure supplement 1.** Specific inhibition of dSPN activity in the DLS through chemogenetics.
DOI: https://doi.org/10.7554/eLife.29908.020
**Figure supplement 2.** Development of action sequences in dSPN-hM4Di(-) and dSPN-hM4Di(+) mice.
DOI: https://doi.org/10.7554/eLife.29908.021

decrease of p-MAPK +neuron density ($t_{(5)}$ = −15.565, p<0.001) in equivalent areas of infected and noninfected hemispheres ($t_{(5)}$ = −1.222, p=0.276).

Based on the territories found to be devoid of p-MAPK +neurons in aged mice (see *Figure 4F*), we induced bilateral transduction of hM4Di DREADD into the DLS in a new cohort of young *Drd1a*-Cre⁺:*Drd2*-eGFP⁺ (dSPN-hM4Di⁽⁺⁾) mice and their littermate controls (*Drd1a*-Cre⁻:*Drd2*-eGFP⁺; dSPN-hM4Di⁽⁻⁾) before LP sequence training (*Figure 5A*). From day 10 onwards, all mice received a 3 mg/kg CNO injection (i.p.) 45 min prior to the start of each session (*Figure 5B*). A two-way mixed ANOVA (factors: training; genotype) showed that the CNO treatment did not affect the ability of mice to acquire instrumental responses, as the rate of LP significantly increased as training progressed in both groups ($F_{(16,192)}$ = 6.389, p<0.001) (*Figure 5—figure supplement 1F*). Importantly, this treatment did not cause a drop in overall performance (*Figure 5—figure supplement 1F*, compare days 9 and 10), which equivalently increased in both groups across training (training x genotype interaction: $F_{(16,192)}$ = 0.601, p=0.881). This suggested that 3 mg/kg of systemic CNO did not cause detectable behavioural side effects.

We studied in these mice whether differences in sequence learning emerged throughout training, similar to the effects observed in aged mice (see *Figure 1*). We found that dSPN-hM4Di⁽⁻⁾ and dSPN-hM4Di⁽⁺⁾ mice produced a similar number of sequences per minute up to day 9, but hM4Di⁽⁺⁾ littermates tended to display higher sequence rates once the CNO treatment started (*Figure 5C*), as supported by a significant between subjects effect in a two-way mixed ANOVA applied only on CNO days ($F_{(1,12)}$ = 6.380, p=0.27). Importantly, both groups showed an equal increase in the number of presses per sequence from day 10 (first day of CNO) to day 17 (*Figure 5D*). Accordingly, a two-way mixed ANOVA showed a significant increase in sequence length in all mice ($F_{(1,12)}$ = 91.036, p<0.001), without a significant day x group interaction ($F_{(1,14)}$ = 19.724, p=0.218). This was further supported by simple effects (factor: day): both dSPN-hM4Di⁽⁻⁾ and dSPN-hM4D⁽⁺⁾ mice significantly increased their sequence lengths ($F_{(1,6)}$ = 59.546, p<0.001; and $F_{(1,6)}$ = 33.521, p<0.01; respectively), indicating that both groups were equally capable of extending (in terms of number of elements) their action sequences. In contrast, we found that only dSPN-hM4Di⁽⁻⁾ controls increased the duration of their sequences from day 10 to 17, whereas dSPN-hM4Di⁽⁺⁾ littermates displayed the same short latencies throughout (*Figure 5E*), as supported by a significant day x genotype interaction ($F_{(1,12)}$ = 5.087, p<0.05). Simple effects confirmed a significant extension of action latency in dSPN-hM4Di⁽⁻⁾ ($F_{(1,6)}$ = 30.545, p<0.01) but not dSPN-hM4Di⁽⁺⁾ mice ($F_{(1,7)}$ = 1.624, p=0.250). Event-time plots recorded in both groups on day 17 revealed that both groups developed clearly defined action sequences (with identifiable initiation, execution and termination elements), although these sequences tended to be more frequent and shorter in dSPN-hM4Di⁽⁺⁾ littermates (*Figure 5—figure supplement 2A–B*).

We next sought to compare the patterns of action chunking displayed by both groups on training day 18, similar to our previous study in young and aged mice (see *Figure 3*). We found that dSPN-hM4Di⁽⁺⁾ mice produced, overall, faster action sequences ($t_{(12)}$ = −2.3841, p=0.035) (*Figure 5F*), and also displayed a higher proportion of short IPIs (*Figure 5G*). This was similar to the effect observed in aged mice (see *Figure 3A–B*), although peak speeds within fast sequence bursts in dSPN-hM4Di⁽⁺⁾ were not as high as the ones recorded in aged mice. Temporal analysis of sequence elements on day 18 revealed that dSPN-hM4Di⁽⁺⁾ mice presented a shifted distribution of within-sequence elements in the interval space (*Figure 5H*) and, consequently, displayed significantly shorter average IPIs ($t_{(12)}$ = 2.964, p=0.012) (*Figure 5J*), which was a similar trend to that observed in aged mice (see *Figure 3J*). However, we found no group differences in the distribution (*Figure 5I*) and average value (*Figure 3K*) of sequence boundary intervals ($t_{(12)}$ = 1.435, p=0.177), which was in contrast to the pattern observed in aged mice (see *Figure 3K*). Altogether, these findings suggest that interrupting intracellular signalling in dSPNs of the DLS during periods of sequence development can increase the speed of individual action sequences, although the number of elements fitted within each sequence as well as the extra sequence intervals remained unaltered. Therefore, this manipulation reproduced within-sequence but not extra-sequence features of action chunking in ageing, a pattern that could explain the increased—rather than reduced—ability of dSPN-hM4Di⁽⁺⁾ mice to obtain rewards during ST training ($t_{(12)}$ = −2.4, p=0.033) (*Figure 5L*).

## Action-related feedback transitorily restores sequence structure in aged mice

A possible source of the behavioural deficit we observed in aged mice is a loss of action-related feedback; the use of environmental feedback in the form of action-associated cues has been reported to improve self-initiated performance in older adults (*Lindenberger and Mayr, 2014*). We, therefore, attempted to normalise the behavioural patterns of aged mice by providing action-related feedback in the form of an external cue. We submitted two groups of aged mice to LP sequence training, both of them receiving a sound cue (0.75 s white noise) that signalled the last LP of each sequence preceding the delivery of reward. For one of the groups the feedback cue was given for the duration of the experiment (group Cue Maintained, days 1–36), whereas for the other group cue presentation stopped on day 19 (group Cue Lost) (*Figure 6A*). This treatment had no apparent effects on the overall performance of either group: two-way mixed ANOVA (factors: training; cue) showed that all mice escalated their instrumental responses throughout the duration of the experiment ($F_{(35,420)}$ = 72.869, p<0.001), and there was no significant training x cue interaction ($F_{(35,420)}$ = 1.378, p=0.078) (*Figure 6B*). We then compared the sequence features displayed on days 7, 18 and 19, the latter being the first day of cue withdrawal for the Cue Lost group (*Figure 6C,D*). Two-way mixed ANOVA (factors: day; cue) showed significant differences in sequence length on days 7, 18 and 19 of training ($F_{(2,26)}$ = 199.993, p<0.001). However, there was a significant day x cue interaction ($F_{(2,26)}$ = 6.567, p<0.01), suggesting that both groups performed differently across these days. Simple effects (factor: day) confirmed that the sequence length dropped significantly from day 18 to day 19 in the Cue Lost group ($F_{(1,8)}$ = 15.696, p<0.01), whereas it remained constant in the Cue Maintained group ($F_{(1,5)}$ = 0.851, p=0.399) (*Figure 6C*). This effect was more pronounced on sequence duration: we found significant differences in the duration of the sequences between days 7, 18 and 19 ($F_{(2,26)}$ = 20.643, p<0.001), and again a significant day x cue interaction ($F_{(2,26)}$ = 6.256, p<0.01) with simple effects confirming that, upon cue removal, the duration of action sequences dropped to pre-sequence levels in group Cue Lost ($F_{(1,8)}$ = 21.432, p<0.01) but not in group Cue Maintained ($F_{(1,5)}$ = 0.996, p=0.364) (*Figure 6D*). These results suggest that the execution of extended action sequences in aged mice depends on the provision of action-related feedback instructing the duration of action.

We next investigated in these mice the transition from long to short sequences on cue withdrawal (day 19). Event-time diagrams representing the sequences executed at the beginning and at the end of the training session showed that all mice in group Cue Lost started by executing long action sequences even in the absence of the auditory cue, but that these sequences rapidly shortened throughout the session (*Figure 6E,F*; *Figure 6—figure supplement 1A,B*). Accordingly, two-way mixed ANOVA (factors: sequence number; period) showed overall significant differences in the length of the sequences as the session progressed ($F_{(9,144)}$ = 3.635, p<0.001). However, there was a significant sequence number x period interaction ($F_{(9,144)}$ = 218,993, p<0.01), indicating that the length was significantly different in the Start and End periods of the session. Simple effects analysis (factor: sequence number) revealed that the length of the sequences progressively decreased at the start of the session ($F_{(9,72)}$ = 5.089, p<0.001), but remained constant towards the end ($F_{(9,72)}$ = 0.921, p=0.512) (*Figure 6F*). Importantly, the shortening of action sequences precipitated by cue removal was translated into a dramatic increase in the total number of sequences expressed by the Cue Lost mice that persisted throughout training (*Figure 6G*), as indicated by a significant day x cue interaction ($F_{(35,420)}$ = 2.339, p<0.001).

Finally, we assessed the action chunking patterns across training in both groups of aged mice and how these patterns changed with cue removal (*Figure 6H*). We found that, despite the presence of the feedback cue, both groups of mice displayed similarly aberrant automaticity on day 10, with boundary intervals accumulating in a narrow band nearby the chunking space (*Figure 6H*, day 10). However, mice seemed to learn to categorise their sequence elements with further cued training as the boundary interval band disappeared in both groups (*Figure 6H*, Day 18). Strikingly, group Cue Lost precipitated microchunks as soon as the instructive cue was removed (*Figure 6H*, compare days 18 and 22). Taken together, these results show that aged mice can execute longer action sequences as long as action-related feedback is provided regarding sequence duration, suggesting that this deficit is cognitive-motor rather than purely executional in origin.

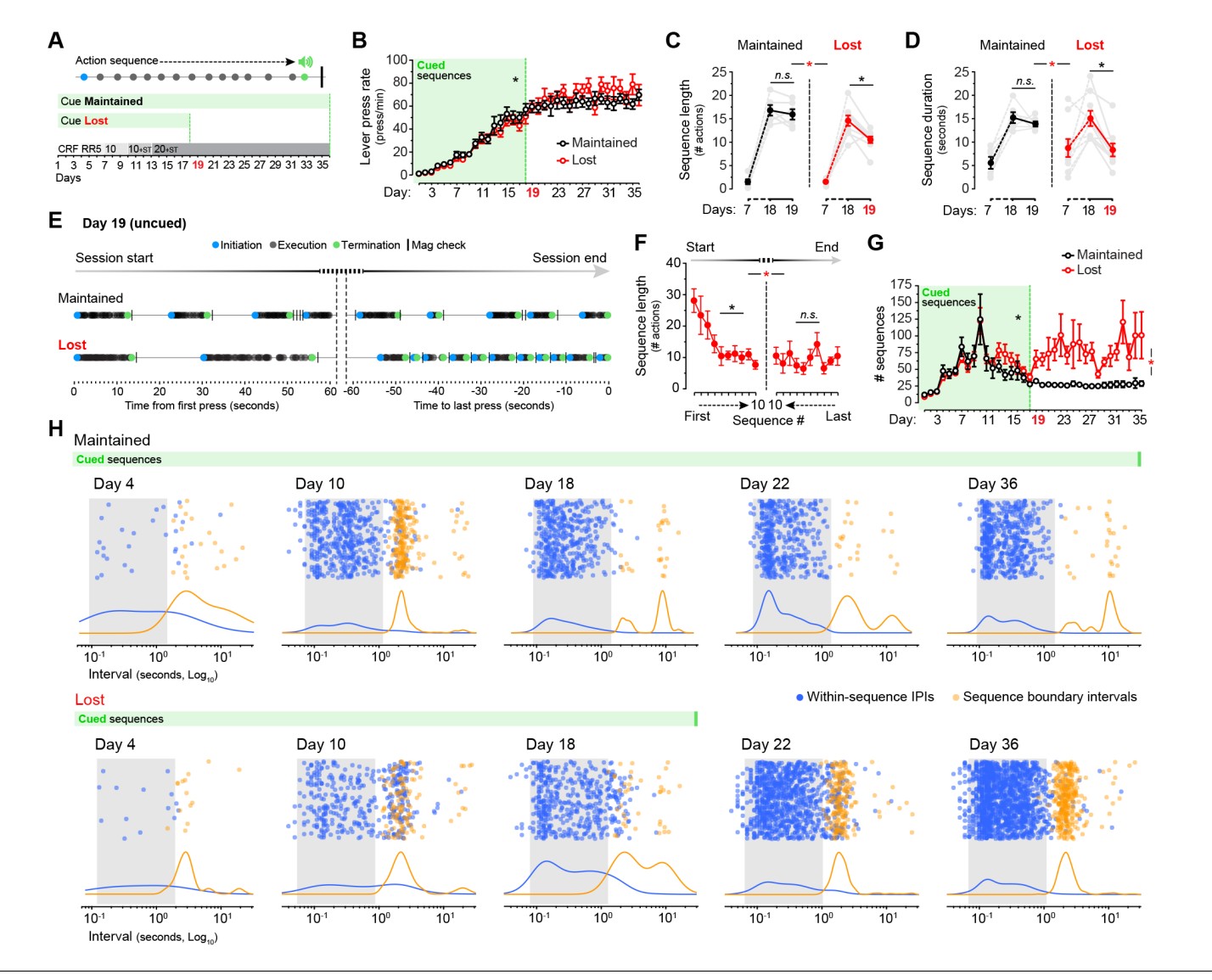

**Figure 6.** Cues signalling sequence termination temporarily normalise action structure in aged mice. [*Figure 6—source data 1*] (A) Aged mice were submitted to extended LP sequence training where the termination element in each sequence was cued with a 0.75 s white noise. One group maintained the termination cue until training day 36 (Cue Maintained), whereas in the other group (Cue Lost) the cue was removed from day 19. (B) Resulting acquisition of LP instrumental behaviour. Data are mean ±SEM (6–8 mice per group). Asterisk denotes significant overall effect of training (see text) [StatsReport18]. (C, D) Average length (C) and duration (D) of the LP sequences produced by both groups on days 7, 18 and 19 of training. Asterisks denote significant day x group interaction (red) and simple effects (black, see text). N.S., not significant [StatsReport19] [StatsReport20]. (E) Structure of LP sequences produced by a 'Cue Maintained' and a 'Cue Lost' mouse at the start and at the end of the session on training day 19. See also *Figure 6—figure supplement 1*. (F) Average length of the 10 first sequences and the 10 last sequences produced by 'Cue Lost' mice on training day 19. Asterisks denote significant sequence x start/end period interaction (red) and simple effects (black, see text). N.S., not significant [StatsReport21]. (G) Total number of action sequences produced in each day of training before and after the loss of the termination cue. Asterisks denote significant overall effect of training (black) and training x group interaction (red, see text) [StatsReport22]. (H) Scatter plots of each within-sequence inter-press-interval (IPIs, blue) and sequence boundary interval (orange) produced by one 'Cue Maintained' (subject #R8_M) and one 'Cue Lost' (subject #B4_L) mouse across different days of training. Kernel density curves of each interval type are plotted at the bottom. Grey shades delimitate the within-sequence chunking space.

DOI: https://doi.org/10.7554/eLife.29908.024

The following source data and figure supplement are available for figure 6:

**Source data 1.** Source data for *Figure 6*.
DOI: https://doi.org/10.7554/eLife.29908.026

**Figure supplement 1.** Aged mice lose action structure as soon as feedback cues are removed.

*Figure 6 continued on next page*

*Figure 6 continued*

DOI: https://doi.org/10.7554/eLife.29908.025

## Discussion

In the present study, we combined a series behavioural designs with comprehensive neuronal mapping and targeted circuit manipulation to reveal a fundamental deficit in the development of action sequences in mice of advanced age (20–22 m.o., average life expectancy is 24 months). It is well known that certain forms of motor skill learning are impaired in older adults (*Voelcker-Rehage, 2008*). In procedural learning, studies in humans demonstrate that both early (65-68) and late elderly (75-88) show very limited use of motor chunking during multi-element sequence tasks (*Shea et al., 2006*; *Verwey, 2010*). Although the precise patterns of action chunking differ between homogenous and heterogeneous sequence learning (i.e. sequences involving one vs. more than one action element) (*Garcia-Colera and Semjen, 1987*; *Sternberg et al., 1978*), our results in mice pressing a single lever revealed that older subjects consistently developed a form of chunking that involved shorter and faster sequences (microchunks, see *Video 2*). Similarly, using a serial response time task in humans, in which the formation of chunks was encouraged using visuospatial cues, *Bo et al. (2009)* found that older adults developed motor chunks overall, although the length of those chunks was reduced. Taken together, these results suggest that the automation of actions is possible in the aged, but is temporally distorted compared to that expressed by younger individuals.

Over and above the task-efficiency benefits of well-developed skills, automated or model-free behaviour has advantages for computational and space complexity loads; operationally, model-free reinforcement learning systems require less memory than model-based systems (*Schneider and Shiffrin, 1977*; *Dayan and Daw, 2008*). One important implication of this is that implementing model-free operations is more appropriate when computational resources such as working memory are limited (*Otto et al., 2013*). On this account, the computational resources of the aged brain can be limited by its clearly reduced working memory capacity (*Bo et al., 2009*; *Wang et al., 2011*; *Wingfield et al., 1988*). Therefore, in light of these memory caching limits, implementing model-free or automatic processes early in training can be seen as a successful strategy. Although this perspective can tentatively explain the formation of premature, unbreakable microchunks during early stages of training in aged mice, there are two features of our data that could argue against it. First, model-based to model-free transition (i.e., goal-directed to habitual modes of action) is thought to rely on shifts from dorsomedial to dorsolateral corticostriatal networks (*Balleine, 2005*; *Yin et al., 2009*). In the particular case of sequence learning and action chunking, engagements of posterolateral corticostriatal circuits have been demonstrated in humans and animals that are producing clearly automatised instrumental behaviours. *Wymbs et al. (2012)* found in humans that activity in the frontoparietal cortex and sensorimotor putamen (equivalent to the posterior dorsolateral striatum in rodents) strongly predicts critical features of action sequence execution, and studies in mice have provided information on the specific circuitry within posterior dorsolateral areas involved in action sequence initiation, execution and termination (*Jin and Costa, 2010*; *Jin et al., 2014*; *Tecuapetla et al., 2016*). Accordingly, a premature implementation of model-free processing (such as observed in microchunking) should predict an early medio-lateral shift of activity in corticostriatal networks. However, our neuronal activity assessments in aged mice suggest the contrary: aged brains seem to fail to disengage rostro-medial corticostriatal circuits, and shorter and faster sequences predict lower activity in dorsolateral regions of the striatum (*Figure 4*). Similarly, preventing such a shift in young mice by disrupting the activity of a subset of neurons in the dorsolateral striatum early in training reproduced some aspects of the aged behaviour, including the shorter latencies and higher speeds (see *Figure 5*).

Secondly, the formation of premature microchunks is not consistent with the results of our last experiment, where similar oligomeric sequences were instantly precipitated by the loss of an action-related feedback cue (*Figure 6*). In this case, aged mice expressed long, uninterrupted sequences from the beginning of training, and never displayed the constricted sequences prior to the loss of the cue. These results suggest that the execution of microchunks is not a consequence of deficient learning, but rather the result of different automaticity processes put in place by the aged brain.

Rather, our study supports the notion that young and aged brains start from a different neuronal environment or scaffolding (*Park and Reuter-Lorenz, 2009*), and attempt to optimise instrumental task performance by implementing entirely different strategies for automaticity. The critical question is: what are the features of the aged brain that produce these shorter patterns of action? Our data suggest that the dynamic medial-to-lateral shifts that are predicted to occur in the corticostriatal network during the development of automatic behaviour (*Yin et al., 2009*) may not happen, or may only occur partially, and this neuronal staticity might be key to explaining the recruitment of unusual automaticity processes in aged mice. Indeed, several functional imaging studies in humans suggest a similar shift in activity to anterior regions of the brain in elderly subjects (see *Park and Reuter-Lorenz, 2009*) for an extensive review): a compensatory overactivation in declining prefrontal cortical regions drives the use of different behavioural strategies to those of younger adults in certain tasks (*Cabeza et al., 2002*; *Gutchess et al., 2005*; *Raz et al., 2005*). This can be regarded as one possible mechanism for the marked differences in motor chunk development in aged humans during sequence learning (*Bo et al., 2009*; *Verwey, 2010*; *Wymbs et al., 2012*).

Nevertheless, why an anterior shift in corticostriatal activity with ageing generates such constricted spurts of action during sequence execution remains unresolved. Again capitalising on age-related working memory deficits and prefrontal cortex dysfunction (*Wang et al., 2011*; *Wingfield et al., 1988*), one could speculate that the overactivity observed in prefrontal areas in aged individuals reflects failed attempts to boost working memory capacity in distal parietal cortical regions, following a similar process to that proposed in computational models of visuospatial working memory retention (*Edin et al., 2009*). Interestingly, our data reveals specific activation reductions in dorsomedial prefrontal cortex as well as frontoparietal cortex, regions likely to be part of the prefronto-parietal network for top-down control of working memory capacity (*Edin et al., 2009*; *Gruber and Goschke, 2004*). Certainly, these prefrontal deficits could contribute to part of the problem, although the question of how working memory contributes to action organisation remains largely unresolved and is complex, likely involving subcortical basal ganglia circuitry (*O'Reilly and Frank, 2006*).

Another, perhaps related, neuronal process that could in part explain the microchunk behaviour in our study is malfunction in the prefronto-basal ganglia network and related deficits in temporal control with ageing (*Gunstad et al., 2006*). One interesting possibility is that the aged brain is subject to constraints during the temporal scaling of sequence processing in corticostriatal networks (*Fukai, 1999*; *Igarashi et al., 2013*). Although the precise mechanism by which time is encoded in the brain is unknown, recent evidence clearly implicates subcortical networks (dorsal striatum) as a key structure in scaling time for action (*Mello et al., 2015*; *Soares et al., 2016*). In our study, we found that selectively reducing the activity of direct pathway spiny projection neurons (dSPNs) in dorsolateral regions of the striatum reduced action duration while increasing action speed. These results directly support and extend recent findings on the involvement of dSPNs in encoding of the latency of action. *Jin et al. (2014)* found that a large proportion of dSPNs recorded in the dorsolateral striatum showed sustained patterns of firing that spanned the duration of an action sequence. More recently, the same group found that optogenetic stimulation of these neurons in the same striatal territory extended the length and duration of ongoing action sequences only during periods of stimulation (*Tecuapetla et al., 2016*). It should be emphasised, however, that although our striatal manipulations may have recapitulated some of the microchunking features observed in aged mice, we did not attempt to recreate accompanying prefronto-parietal dysfunction (and the resulting working memory deficit). This might explain why our young mice with the striatal manipulation were still capable of building up a reasonable action structure, something that unexpectedly rendered the animals more efficient at obtaining outcomes (*Figure 5*). Thus, our results add to studies elucidating the microcircuitry responsible for the temporal scaling of action, and suggest the intriguing possibility that the static properties of aged cortico-basal ganglia networks have direct effects on the duration of action sequences, compromising behavioural automaticity.

In conclusion, we provide evidence for a severe deficit in corticostriatal function during normal ageing in mice that engenders a distortion of action sequences after extended training. Our findings have implications for skill learning in ageing and point to the stationarity of cortico-basal ganglia transmission as a major feature of the aged brain hampering its ability to acquire new skills. Importantly, we found that action-related feedback cues given throughout training transitorily solves this

problem, a solution that could be readily implemented in current rehabilitation strategies and behavioural therapies for cognitive ageing (*Lindenberger and Mayr, 2014*).

## Materials and methods

### Mice

All procedures were approved by the University of Queensland Animal Ethics Committee (QBI/412/14/NHMRC and QBI/027/12/NHMRC) in accordance with the *Animal Care and Protection Regulation* (Queensland Government, 2012)and the *Australian Code for the Care and Use of Animals for Scientific Purposes* (*National Health and Medical Research Council, 2013*National Health and Medical Research Council, 2013).

C57BL6/J mice (RRID:IMSR_JAX:000664) were purchased from the Animal Resources Centre (Perth, Australia), and an ageing colony was established in the Queensland Brain Institute Animal Facility. Young mice used for the experiments were 2 months old at the beginning of the experiments, whereas aged mice were 20–22 months old. Transgenic mice were purchased from the Mutant Mouse Resource and Research Centers (MMRRC, NIH) and outbred to C57BL6/J (RRID: IMSR_JAX:000664) for more than six generations in the Queensland Brain Institute Animal Facility. Heterozygous *Drd1a*-Cre$^{+/-}$ mice (strain B6(Cg)-Tg(Drd1a-cre)FK150Gsat/Mmucd [RRID:MMRRC_029178-UCD]: expresses Cre in dSPNs) were bred with homozygous *Drd2*-EGFP mice (strain B6(Cg)-Tg(Drd2-EGFP)S118Gsat/KreMmucd [RRID:MMRRC_036931-UCD]: expresses EGFP in iSPNs). This crossing produced 50% of *Drd1a*-Cre$^{+/-}$ *Drd2*-eGFP$^{+/-}$ mice and 50% of *Drd1a*-Cre$^{-/-}$ *Drd2*-eGFP$^{+/-}$ mice, which were used in the experiments at 2 months of age. All mice weighted 23–26 g (young) and 28–34 g (aged) at the beginning of each experiment, although food intake was restricted throughout most of the experiments (mice were maintained at approximately 85% of their free-feeding weight). All mice were housed in plastic home cages (Optimice, Animal Care Systems, CO) in a climate-controlled colony room on a 12 hr light/dark cycle and handled daily, and were allowed *ad libitum* access to water. Littermates in each home cage were maintained throughout the experiment, and were randomly allocated to each experimental group.

### Behavioural procedures

#### Apparatus

All training procedures were carried out in sound- and light-resistant operant chambers (MED Associates, PO). The chambers were illuminated with a 3 W, 24 V house light. Each chamber contained a recessed feeding magazine in the center of the right side wall connected to two individual food dispensers that could deliver 20 mg dustless precision pellets (Bio Serv, NJ, #F0163) into the magazine when activated. Either side of the magazine contained retractable levers. Med-PC software was used to direct the insertion and retraction of the levers, illumination of the light and delivery of the pellets. The software allowed the recording of timestamp events each 10 msec, including the number of lever presses, magazine entries and food pellets delivered throughout the duration of the experiment. Activity was also monitored using D-Link DCS-932L IP LED infrared cameras (VGA 1/5 inch CMOS sensor with built-in microphone) through D-ViewCam Software (D-link Corporation, Taiwan).

#### Magazine training

Prior to all instrumental procedures, mice were exposed to four or six days of magazine training in the operant chambers (one session/day). Each mouse was assigned one operant chamber, which was maintained throughout the duration of the experiment. The chamber light was illuminated to signal the beginning of the session and extinguished at its termination. Both levers were retracted and mice explored the chamber freely. Twenty grain food pellets (20 mg, 3.35 kcal/g each) were delivered into the magazine at random time intervals (RT 60 s) over ~30 min. These pellets differed in taste but were calorically similar (grain: 3.35 kcal/g, purified: 3.6 kcal/g). Consumption of pellets was monitored at the end of each session.

## Standard instrumental training

All behavioural procedures were based on single-lever operant conditioning and started with a standard instrumental training module. Each training session began with the illumination of the chamber light and insertion of the two levers into the operant box, and ended with the extinguishing of the chamber light and lever retraction. The mice were exposed to one training session each day where only one of the levers was contingent to the delivery of the pellets (active lever). Assignment of right or left active lever was counterbalanced across mice. Each training session continued until mice had received a total of 20 pellets, or the session timed out at 30 min. For the first three days of instrumental training (days 1–3), mice were trained on a constant reinforcement schedule (CRF); every lever pressing action was rewarded. The probability of the outcome given a response (P) was gradually shifted over the following days of training using increasing random ratio (RR) schedules of reinforcement: a RR5 schedule (p=0.2) was always used on days 4–6, and RR10 (p=0.1) and RR20 (p=0.05) were applied from day seven as indicated in each particular experiment (see below).

## Sequence training

Sequence training procedures started with standard instrumental training (days 1–6). RR10 and RR20 were then applied as indicated in each experiment (see *Figures 1A*, *5E* and *6A*). In parallel to this schedule of reinforcement, an additional rule to promote sequence learning was applied from day 10 onwards (sequence trigger, ST, *Figure 1—figure supplement 1*), which limited the access to RR reinforcement to action sequences that were initiated with at least 5 (applied from day 10) or 7 (applied from day 13) consecutive lever presses. Sequences that were truncated with a termination element (lever press → magazine entry) before reaching the fifth or seventh consecutive press blocked access to the RR program. Unlocked sequences (≥5 initial presses) that were truncated with a termination element before earning reward also reset the RR trial. This was achieved in MED PC-IV by caching and verifying, for each lever press, the identity of the behavioural elements stamped 5 or 7 positions back in the timestamp array along each round of reinforcement. On training day 18, the mice underwent an additional training session that lasted for 20 min and had no limit of rewards prior to perfusion (see section 'Transcardial fixation and tissue sectioning' below). In the feedback cue experiment (*Figure 6*), the termination element in each sequence (i.e., the lever press that triggered the delivery of the reward) was cued with a 0.75 s white noise throughout training (see *Figure 6A*). On day 19, the cohort of mice was divided in two groups, according to matching performances. Group Maintained kept the same procedure with cued sequences until day 36, whereas Group Lost stopped receiving such cue but kept all the other parameters of the task unchanged. Mice in Group Lost were run in a separate session to prevent sound propagation from cued animals.

## LH training

Lever Hold (LH) operant conditioning procedures (*Figure 2*) were a variation of our previously reported variable interval hold task (*Bailey et al., 2015*), and started with standard instrumental training (days 1–3), where each individual press was reinforced (CRF or LH0). The probability of outcome delivery was then rendered dependent on the time of lever hold, and was gradually shifted over the following days of training using increasing pseudorandom lever hold (LH0.3-LH5) intervals. During LH0.3 (days 4–6), the distributions of required hold durations had a mean of ~0.3 s (LH0.3: min = 0.05 s; max = 0.88 s). The required hold durations for the subsequent sessions were drawn from an exponential distribution with a higher mean (LH0.8 [days 7–9], LH2 [days 10–12], LH3.2 [days 13–15] and LH5 [days 16–17]). Thus, during the final session of LH training, subjects were required to hold down the lever for intervals that averaged 5 s but could be as long as 11.7 s.

## Chemogenetics

Specific inhibition of dSPNs regionally contained in the DLS was achieved by pulse-based microinjection of Cre-dependent hM4Di DREADDs virus (AAV2-hSyn-DIO-hM4D(Gi)-mCherry, UNC Vector Core, NC) into the DLS at two different rostro-caudal levels of *Drd1a*-Cre$^{-/+}$ *Drd2*-EGFP$^{+/-}$ and *Drd1a*-Cre$^{-/-}$ *Drd2*-EGFP$^{+/-}$ double transgenic mice through stereotaxic surgery (coordinates: R-C: +1.1/+0 mm [bregma]; M-L: ±2/±2.75 mm [bregma]; D-V: −2.625/–2.75 mm [skull surface]) combined with an intraperitoneal injection of clozapine N-oxyde (CNO; dissolved in saline to 3 mg/kg; NIMH Chemical Synthesis and Drug Supply Program, Bethesda, MD). In order to validate the effects of this

manipulation, a first cohort of these mice (2 Drd1a-Cre$^{-/+}$ Drd2-EGFP$^{+/-}$ and 2 Drd1a-Cre$^{-/-}$ Drd2-EGFP$^{+/-}$) was used where the virus was injected unilaterally (see **Figure 5A**). After recovery from surgery, these mice received a first injection of CNO and 30 min later another injection of GBR12783 (15 mg/kg, i.p. dissolved in sterile water) prior to rapid intracardial perfusion (see **Figure 5A–C**). Animals used for sequence training received a bilateral injection of the virus and were injected with CNO on days 10–17 45 min prior to the start of the training session (see **Figure 5D–L**).

## Stereotaxic surgery

Mice were pre-anesthetised in an induction chamber with oxygen/isoflurane mixture (Laser Animal Health, Pharmachem, Australia). Pedal reflex was used to monitor anesthesia before placing the mouse in the stereotaxic frame (Kopf Instruments) fitted with a mask supplying a continuous flow of oxygen/isoflurane mixture (1 L/min oxygen, 1.5 ml/min isoflurane). Once deeply anesthetised, a 1.5 cm incision was made to expose the skull. For each microinjection, a small hole was pierced (~0.2 mm width) at the appropriate A-P and M-L coordinates (see above). A pulled glass capillary (GC100TF-15, Harvard Apparatus) was pulled using a micropipette puller (P-97, Sutter Instrument) and pre-filled with the solution and fitted into a Nanoject II (Drummond Scientific) before being slowly inserted vertically through each hole until reaching target (D-V coordinate). Once in target, injections were preceded by a 2 min waiting interval. Two sets of pulse injections of hM4Di-DREADDs AAV were delivered in one (unilateral experiments) or both (bilateral experiments) sides of the brain. Each injection consisted of 14 pulses of 69 nL spaced by 30 s. Two min after the last pulse, the pipette was gently pulled out, and the incision was closed with surgery suture silk and sealed with tissue adhesive (3M Vetbond). Animals were placed in cages separately on a warming plate (37°C) during recovery. A pre-operative s.c. administration of butorphanol (torbugesic, 3 mg/kg) ensured analgesia during and after the intervention. The antibiotic Baytril (5 mg/Kg) was administered at the end of the procedure. Both torbugesic and baytril were administered once a day for 2 days after surgery. Behavioural procedures started after a 3 week recovery period. None of the mice displayed perceptible wounds by the start of the behavioural experiments.

## Tissue processing and immunofluorescence

### Transcardial fixation and tissue sectioning

Twenty min after the start of the session (day 18), the mice were rapidly anesthetised by exposure to 10 s of isoflurane gas (4% in air; Laser Animal Health, Pharmachem, Australia) in a sealed chamber, followed by a lethal intraperitoneal injection of sodium pentobarbital (500 mg/kg; Virbac Pty. Ltd., Australia). After ensuring deep anesthesia by testing paw and tail reflexes, mice were perfused transcardially using an air pressure system (constant flow of 15 ml/min) with 4% paraformaldehyde (PFA) in a solution of 0.1 M sodium phosphate buffer (pH 7.4). Brains were dissected and post-fixed overnight in PFA solution at 4°C. Consecutive 30 µm coronal sections spanning the rostro-caudal extent of the striatum were obtained for each animal using a vibratome (VT1000s, Leica Microsystems, Germany). Free-floating slices were stored at −20°C in a cryoprotectant solution (30% ethylene glycol, 30% glycerol, 0.1 M sodium phosphate buffer) until processed for immunofluorescence.

### Immunofluorescence

Free-floating slices were rinsed three times in Tris-buffered saline (with sodium fluoride id phosphorylations were detected: TBS NaF; 0.25 M Tris, 0.5 M NaCl, 0.1 mM NaF, pH 7.5) for 10 min on an orbital shaker at room temperature prior to membrane permeabilisation treatment (0.3% triton X-100 in TBS [NaF]), which was applied for 1 hr. After three 10 min washes with TBS (NaF), slices were incubated at 4°C on an orbital shaker with a rabbit polyclonal anti- threonine 202 and tyrosine 204-phosphorylated p44/42 MAPK-ERK1/2 primary antibody (p-MAPK; diluted 1:500; Cell Signaling Technology, MA; Cat# 9101L RRID:AB_331646). Following a 24 hr incubation period, unbound primary antibodies were washed off with 3 washes of TBS (NaF), and bound primary antibodies were detected through incubation for 1 hr at room temperature with donkey anti-rabbit fluorescent Cy3-conjugated secondary antibodies (diluted 1:800, Jackson ImmunoResearch Laboratories, PA). In the viral infection studies, simultaneous detection of iSPN fluorescence (Drd2-eGFP), dSPN virus expression (Drd1a-hM4Di-mCherry) and neuronal activation (p-MAPK, Cy5 donkey anti-rabbit antibody, 1:800 dilution) was conducted. Unbound secondary antibodies were washed off through four final

rinses in TBS. Samples were then placed on Superfrost Plus coated slides (Thermo Fisher Scientific, MA) and mounted with a coverslip on Vectashield fluorescence medium with or without DAPI nuclear staining (Vector Laboratories, CA). Slides were stored at 4°C in the dark until image acquisition.

## Image acquisition

All image datasets in this study were obtained in the Queensland Brain Institute Advanced Microimaging and Analysis Facility.

### Spinning disk microscopy

Neuronal activity mapping experiments were performed through wide-field spinning disk confocal microscopy. A spinning disk confocal microscope (W1 Yokogawa spinning disk module; Zeiss Axio Observer Z1) equipped with sCMOS camera (Hamamatsu Flash 4.0 with 2048 × 2048 chip) was used to capture p-MAPK-labeled individual tiles (optical magnification: 20X; pixel depth: 16 bit; image resolution: 0.625 pixels/μm; image size: 1024 × 1024 pixels, 561 laser). Tiles were stitched together using Slidebook 6.0 software to reconstruct a high-resolution image of bilateral striata. In each mouse, three sections at the following approximate coordinates were reconstructed (A-P distance from bregma): 1.7 mm (rostral, 1.1 mm (medial) and 0.15 mm (caudal). Three-channel tiles (488 nm, 561 nm and 637 nm lasers) were obtained in viral infection experiments.

### Scanning confocal microscopy

Simultaneous analysis of iSPN eGFP fluorescence, hM4Di-mCherry virus expression and p-MAPK distribution was performed through point scanning confocal microscopy. Single plane confocal images (optical magnification: 40X; averaging scans: 4; pixel depth: 12-bit) were captured using a Zeiss LSM 710 confocal laser-scanning microscope (Carl Zeiss AG, Oberkochen, Germany). Three-channel images (488 nm, 561 nm and 633 nm lasers) of the dorsolateral striatum were taken for each hemisphere in 2 *Drd1a*-Cre$^{-/+}$ *Drd2*-EGFP$^{+/-}$ and 2 *Drd1a*-Cre$^{-/-}$ *Drd2*-EGFP$^{+/-}$ mice.

## Data acquisition and analysis

### Action data (timestamp)

[*Figure 1—source data 1*, *Figure 2—source data 1*, *Figure 5—source data 1*, *Figure 6—source data 1*, *Figure 1—source data 3*] A combination of MEDState Notation (MSN, MED Associates) language and a series of customised MATLAB (MathWorks, Natick, MA) scripts were used to retrieve, sort and plot timestamp datasets. All individual events (left and right lever press, magazine entries, and pellet deliveries) that occurred during each training session were recorded. The time at which each event occurred (y matrix) was stamped with a 10 ms resolution (z matrix, centiseconds). On each subject, streams of timestamp data of each category (i.e. 1–4) were combined and chronologically organised. Discrete sequences and their initiation, execution and termination elements were then identified based on the interspacing of lever presses and magazine checks. Consecutive magazine checks produced after the termination of a sequence were ignored. Parameters such as within-sequence inter-press-intervals (IPIs), sequence length (number of actions per sequence) and sequence latency (time elapsed from initiation to termination) were calculated at this point for each sequence, and average values were obtained for each mouse. Chunking during action sequence learning was represented with scatter plots of identified sequence elements (see *Figure 3G,J,K*; *Figure 3—figure supplement 2*; *Figure 5H*; *Figure 6H*). Action intervals were categorised into within-sequence IPIs and sequence boundary intervals. Within-sequence IPIs (execution) involved the intervals formed between presses contained within each sequence. Sequence boundary intervals were calculated by adding up press-check (termination) and check-press (initiation) intervals. Boundary intervals that went beyond 20 s involved engagement in other behaviours and were disregarded. Data in the scatter plots are random (y axis) and Log10 transformation of element intervals (x axis). Kernel density curves indicating clustering of each interval type within a particular session are also plotted.

### Action data (acoustics)

[Figure 3—source data 3] Due to the temporal bias induced by timestamp-based data retrieval at short time intervals (see *Figure 3—figure supplement 1C*), we calculated real inter press interval

times on selected fast sequences on day 18 (see *Figure 3A–C*). The five fastest sequences (from the timestamp data) were identified in the video recordings (.asf files) in each mouse, and the corresponding audio file (.wav) was stripped using VideoLAN software (VLC v2.1.3). Bursts of lever presses were isolated by identifying the acoustic fingerprint of single lever depressions based on superimposing the waveform frequency trace (MATLAB) with the matching spectrogram (WaveSurfer software) (see *Figure 3C*). This method allowed to confidently distinguishing discrete lever press sounds over the background noise as well as to rule out lever-bouncing artefacts and other unrelated sounds. Isolated sequences were confirmed by video inspection. Real IPI and speed data were then calculated in each burst by measuring the time between and frequency of peaks (see *Figure 3C*).

## Pellet drop detection

[SourceData 7]. Recordings were performed inside the same mouse operant conditioning chambers used in all experiments (MED Associates, PO), with all doors closed and the fans activated. Ten food pellets were randomly delivered to the food receptacle by activating the pellet dispenser output (electric rotor moves 1/12th of a turn) with a p=0.1 each second. Sessions ended when 10 pellets had been delivered (average session time = 1.66 min). Pellet deliveries were timestamped with 1 ms resolution. During this time, parallel sound level and vibration recordings were conducted in different devices. Sound level data was recorded using Sound Analyzer v2.2 at variable time weightings (Fast [F]: 125 ms up and down; Slow [S]: 1 s up and down; and Impulse [I]: 35 ms up and 1.5 s down) and different frequency filters (Z [zero]-filter: audio range 10 Hz to 20 kHz ±1.5 dB; A-filter: audio range −20 Hz to 20 kHz; C-filter: loud sound detection). Sound exposure level (E) and equivalent continuous sound level (eq) were also recorded for each filter. Only recordings best detecting the pellet drop sound are shown in *Figure 1—figure supplement 1A*. Vibration was recorded with Vib-Sensor v2.1.1 (Now Instruments and Software) as high-pass filtered accelerometer data (99.3 Hz) on three different movement axes (x, y, z).

## Phospho-MAPK mapping

[*Figure 4—source data 1*] High-resolution mosaics obtained in Slidebook 6.0 (3 per animal from rostral, medial and caudal sections; see 'Image acquisition' section above) were processed using the non-commercial open-source ImageJ2/Fiji software (v.1.49j10; Wayne Rasband, National Institutes of Health, USA). The following cortical and striatal regions were analysed: Prelimbic cortex [PrL], anterior cingulate/Motor region 2 [aCg/M2], anterior dorsal striatum [aDS], core [Co], shell [Sh], dorsomedial striatum [DMS], dorsolateral striatum [DLS], frontoparietal cortex [FPx]. The outer boundary of the different cortical and striatal regions was manually drawn in each hemisection, the area of the region was measured ($mm^2$), and Cartesian coordinates ($x$, $y$) of the points defining the edge were obtained. To quantify the number of activated neurons and retrieve their location across the striatum, images were thresholded based on p-MAPK immunolabelling and individual somata were automatically detected using the 'Analyze Particle' command in ImageJ2/Fiji, which finds the edge of an object, determines its position and outlines its contour creating a region of interest (ROI). Resulting data, including neuron count, position (x, y coordinates) and area ($px^2$) of the centroid of each labeled neuron, was imported into MATLAB (MathWorks, Natick, MA). We then used a customised script that performs the in-built 'inpolygon' function, which returns a list of the points lying inside the edge of the polygon area (cortical and striatal regions). The spatial distribution of neurons was digitally reconstructed by combining a line plot defined by the edge points of the regions and a scatter plot with circles at the locations specified by the neurons' centroid coordinates.

## Statistics

Data were analysed using SPSS Statistics software (version 22; IBM Corporation, Somers, NY). The *a priori* alpha level was set at p<0.05. All statistical details of each experiment, including the n, the statistical test applied and significance, can be found in the Results and in the figure and figure legends. The factors involved are indicated in the text (in brackets). To analyse the effect of one or more independent variables, Levene's test was first used to test the null hypothesis that the error variance of each dependent variable was equal across groups, and uni or multi-factorial ANOVA was then conducted. If one or more repeatedly measured variables were considered, uni- or

multifactorial repeated measures ANOVA was conducted. Homogeneity of variance was tested through Mauchly's test of sphericity, and Greenhouse-Geisser corrections were considered when sphericity was not met. In mixed ANOVA analyses involving within-subject and between subjects comparisons, factors in brackets corresponded to within-subject (first) and between subjects (second). In instances where further statistical detail was appropriate, we conducted additional simple effects comparisons based on individual one-way ANOVAs. Statistical comparisons between two means were conducted through independent t-tests, and equal variances were assumed or not according to the Levene's test for equality of variances. To measure the relationship between two variables, Pearson's correlation coefficient (r) was calculated, where the value can lie between $-1$ (as one variable changes, the other changes proportionally in the opposite direction), to $+1$ (as one variable changes, the other changes in the same direction by the same amount). Correlations involved within-subjects measures.

## Acknowledgements

The authors thank Dr Amir Dezfouli and Prof Mark Bouton for insightful discussions, and Dr Richard Faville and Matthew Van De Poll for assistance in MATLAB programming.

## Additional information

### Funding

| Funder | Grant reference number | Author |
| --- | --- | --- |
| Australian Research Council | DE160101275 | Jesus Bertran-Gonzalez |
| Australian Research Council | DP130101932 | Jürgen Götz |
| National Health and Medical Research Council | APP1037746 | Jürgen Götz |
| National Health and Medical Research Council | APP1003150 | Jürgen Götz |
| National Health and Medical Research Council | GNT1079561 | Bernard W Balleine |

The funders had no role in study design, data collection and interpretation, or the decision to submit the work for publication.

### Author contributions

Miriam Matamales, Conceptualization, Data curation, Software, Supervision, Validation, Investigation, Methodology, Writing—original draft; Zala Skrbis, Data curation, Formal analysis, Investigation, Visualization, Methodology, Writing—review and editing; Matthew R Bailey, Peter D Balsam, Conceptualization, Software, Investigation, Visualization, Methodology, Writing—review and editing; Bernard W Balleine, Conceptualization, Resources, Supervision, Funding acquisition, Investigation, Methodology, Writing—review and editing; Jürgen Götz, Conceptualization, Resources, Funding acquisition, Validation, Writing—review and editing; Jesus Bertran-Gonzalez, Conceptualization, Data curation, Formal analysis, Funding acquisition, Investigation, Methodology, Writing—original draft, Project administration, Writing—review and editing

### Author ORCIDs

Miriam Matamales http://orcid.org/0000-0001-9978-0091
Bernard W Balleine http://orcid.org/0000-0001-8618-7950
Jesus Bertran-Gonzalez http://orcid.org/0000-0002-3794-1782

### Ethics

Animal experimentation: All procedures were approved by the University of Queensland Animal Ethics Committee (QBI/412/14/NHMRC and QBI/027/12/NHMRC) in accordance with the Animal Care and Protection Regulation (Queensland Government, 2012) and the Australian Code of Practice

for the Care and Use of Animals for Scientific Purposes (National Health and Medical Research Council, 2013). All surgery was performed under isoflurane gas anesthesia and butorphanol analgesia, and every effort was made to minimize suffering.

## Decision letter and Author response
Decision letter https://doi.org/10.7554/eLife.29908.029
Author response https://doi.org/10.7554/eLife.29908.030

## Additional files

### Supplementary files
• Supplementary file 1. Stats reports.
DOI: https://doi.org/10.7554/eLife.29908.027

• Transparent reporting form
DOI: https://doi.org/10.7554/eLife.29908.028

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
