## [Decision Letter]

Thank you for submitting your article "Corticostriatal deficit introduces temporal limits to automatic action in aging" for consideration by *eLife*. Your article has been reviewed by three peer reviewers, one of whom, Geoffrey Schoenbaum (Reviewer#1), is a member of our Board of Reviewing Editors, and the evaluation has been overseen by Richard Ivry as the Senior Editor.

The reviewers have discussed the reviews with one another and the Reviewing Editor has drafted this decision to help you prepare a revised submission.

Summary:

In this study the authors tested the ability of young and aged mice to organize actions in an RR20 lever pressing task into discrete sequences or chunks. Young controls developed lever press sequences that were of the approximate correct length to obtain reward efficiently, with short periods between presses, and longer periods preceding the first press after a magazine entry or following the last press before entry. Aged mice also increased their pressing, however their behavior was more sporadic. Sequences were shorter, executed much more quickly, and gaps between presses and other presses and between presses and food magazine behavior were not distinguishable. As a result, aged mice were much less efficient at harvesting reward. The authors use MAPK expression to identify a circuit including mPFC and dorsal striatum normally involved in habit learning in which activity seemed to be reduced in aged subjects. Cuing aged rats with a tone at the end of the sequences allowed for temporary rescue of their behavior, indicating changes were not low level motor problems. But improved performance was not lasting if the cue was removed. Chemogenetic inactivation of direct pathway neurons in young controls in dorsolateral striatum reproduced some of the speeding and microchunking behavior but did not affect overall efficiency the same way.

Essential revisions:

The reviewers identified 3 areas that required revisions that were deemed essential for publication. The first and most important is to clarify that the behavior shown by young rats is in fact chunked and that this is lost in aged rats. While the current detailed analysis was appreciated by all 3 reviewers, it lacks some of the traditional measures to show this. This is captured most clearly by the comment by reviewer 2 on sequence variability in Figure 1 and also analysis of Figure 3. The second critical area was to address whether the deficit shown is primarily in sequence generation or whether this is secondary to another problem, such as inability to hear the pellet dispenser or differences in the use of the trigger in the sequence in aged rats or perhaps simply locomotor problems. Finally there were concerns about the dose of CNO used and also generally the differences between the pattern of effect of the deficit in this experiment versus what is shown by the aged rats. These must all be addressed to satisfy the reviewers.

Reviewer #1:

In this study the authors tested the ability of young and aged mice to organize actions in a simple lever pressing task in to discrete sequences or chunks. Mice were trained on a RR20 task in which chunking was encouraged by a requirement that a series of presses be executed without checking the magazine in order to "access" the RR20 sequence. In young controls, this resulted in the development of lever press sequences that were of the approximate correct length, with short periods between presses, and longer periods preceding the first press after a magazine entry or following the last press before entry. Aged mice also increased their pressing, however their behavior was more sporadic. Sequences were shorter, executed much more quickly, and gaps between presses and other presses and between presses and food magazine behavior were not distinguishable. As a result, aged mice were much less efficient at harvesting reward. The authors use MAPK expression to identify a circuit including mPFC and dorsal striatum normally involved in habit learning in which activity seemed to be reduced in aged subjects. Cuing aged rats with a tone at the end of the sequences allowed for temporary rescue of their behavior, indicating changes were not low level motor problems. But improved performance was not lasting if the cue was removed. Chemogenetic inactivation of direct pathway neurons in young controls in dorsolateral striatum reproduced some of the speeding and microchunking behavior but did not affect overall efficiency the same way.

Overall the study is excellent. It addresses a very interesting question of how action chunking is affected by aging with an elegant and carefully designed and analyzed task. The authors use gene expression to link changes in aged mice with alterations in a neural circuit implicated by a number of labs in the motor sequence learning, and they go the extra mile to manipulate an important part of this circuit to reproduce part of the deficit which gives confidence that the deficits are related to the genetic markers of function. I only have a few suggestions.

One question I had is whether there is any relationship between the aged rats behavior and the unusual requirement for a short burst of presses to access the RR20 routine. I am not sure I understand fully what this means and its implications, but it seemed to me initially that the microchunks could be influenced by this unusual requirement. Is there any possibility of this?

Second I think the authors need to do a better job clarifying the relationship they see between the aging deficit and the effects of the chemogenetic manipulation. It reproduced some features of the aged mice, but it did not cause the loss of efficiency or (oddly) the pattern of IPI's between lever presses versus magazine-lever interactions (panels K). This seems like it deserves to be noted more clearly in the results (assuming I am not misunderstanding) and hit more head on in the discussion. Obviously there is no reason to require one specific pathway to account for all the changes in aging. Just saying this more clearly would be helpful, unless there is a more specific or different explanation.

Lastly I think the authors should make clearer in the discussion why one would think the microchunking in the aged mice is a form of habit or model-free learning. They discuss nicely why their data suggests it is not. But there was not much of a lead in to this section to convince me this was an important question.

Reviewer #2:

This is an interesting body of work examining the capacity for sequence generation in aged mice. A self-initiated paradigm is employed (minus Figure 6) to examine the generation of sequences in lever press behavior devoid of any obvious predictive cue. They report decreased capacity of aged mice to develop automatic sequences, and proceed to investigate circuit activation, potential contribution, and rescue. While there are some nice behavioral analyses, it is not clear that mice are producing automatic sequences; there is a noted absence of analyses on the variability in any aspect of sequence generation. There are major concerns about the circuit activity analyses, and the targeted circuit manipulation was a blunt measure of direct pathway involvement and does not provide any insight into how D1 SPNs may differentially encode or represent sequences. In addition, there are interpretational concerns that arise with the data from the rescue experiment. Overall, while the data approach was interesting, it largely appears that older mice might just be checking more, or at least aging predisposes an increase in checking behaviors. At this point, the conclusion that aging results in impairments specifically to sequence generation in aged mice and related deficits in associated neurobiological mechanisms is not all together convincing.

Concerns

Figure 1: In relation to Figure 1, necessary data and analyses for interpretation are missing. Do young and old mice differ on the amount of sequences performed? Data should be included on how well mice were able to meet the sequence trigger (the efficiency graph on Figure 3 addresses overall performance, but not performance of the ST). It is possible that the aged mice never learned the contingency between the ST and access to the RR, particularly as age is known to engender behavioral inflexibility, and pre-training without the ST requirement may have prevented its acquisition. It would similarly be of interest to examine sequence behavior in these two groups prior to the introduction of the ST requirement.

In addition, the authors should include analysis and discussion of how variable their sequence behavior is. In Figure 1 it appears as though, even by day 17 of training, there is substantial variability associated with the sequences length and duration, both within and across subjects. To what extent are the mice actually performing stereotyped sequences vs. variable ones? This is an important concern, since automaticity would be expected to lead to relatively invariable sequences, yet it appears as though the sequences are highly variable, and therefore, it is unclear how mice are understanding and solving the task.

Also missing are appropriate post hoc analyses that would arise from the significant interactions reported for D and E, which may affect the conclusions proposed. For example, a significant interaction is reported for Figure 1, and indeed the magnitude of sequence length is visually different between young and aged. However, all additional follow up analyses were not performed. They report a main factor of day for both young and old, but no comparison is made between young and aged for the different days. It seems likely that a difference would arise between ages on Day 17. This could then raise an additional hypothesis that aged mice do not extend sequences in the same manner, and importantly, and such, changes in the variability of such sequences needs to be examined.

Figure 3: The chunking analyses are quite interesting, however, they rely on visual interpretation of differences. The authors should apply some quantitative methods to these data to assess group differences. For instance, visually (figure supplement) it appears as though aged mice simply perform many more sequences than young mice (see concern above about amount of sequences), which might explain some of the chunking results. Proportional analyses may be appropriate here. As it stands, this data does not provide evidence that aged and young animals chunk sequences differently (and may suggest different amounts of overall lever pressing). Average inter-event intervals of various types could also be compared and would be informative.

Figure 4: The investigation into how recruitment of activity in corticostriatal networks represented in Figure 4 is somewhat troubling, in that it may be inherently confounded by general differences in lever pressing and have nothing to do with sequence generation. Are the groups matched for total lever presses? Is it that one group (aged) is just making more lever presses and hence, that could account for the differential activation in the circuit (based on action execution and nothing to do with sequence generation)? In Figure 4, the large variability between subjects is apparent, and while various sequence factors were correlated with counts observed, whether overall lever presses correlated was not presented. Given this variability within a given group, do similar relationships arise for within group correlations? Further, for the analyses in D and E, how is the variability in each animal factor into the variability in the overall analyses? This is not clear in the Materials and methods. Are total activity levels normalized is some manner?

Figure 5: The CNO dosage used is quite high (3mg/kg), and this coupled with repeated dosing across days raises concerns about the potential changes in plasticity that may have occurred. This would raise alternative hypotheses other than an effect of altered D1 SPNs underlying sequence deficits in older animals. If specifically, D1 SPNs control over sequence execution is altered in aged mice, then acute attenuation should result in some alteration in performance. Also, it is unclear from the present data if dMSNs in the DLS are important in learning/acquiring these sequences, in their performance, or both. Although not unconvincing, for Figure 5, the authors should have consistency between the data presented here and in Figure 3 and C. In Figure 3, the authors show IPI and speed data from only the fastest sequences, while in Figure 5 the speed and IPI data appears to be collapsed across all sequences. Does CNO replicate the effect on the fastest sequences, and additionally, does age have the same overall effect as CNO appears to? Is there any effect on the overall number of sequences? Why was the comparison made at day 10 instead of 7 (different schedule requirements were in place)? Perhaps the authors should temper their comparisons between age and dMSN inhibition. Particularly since CNO did not recapitulate the effects of age on either action chunking or task performance (where indeed, CNO and age had opposite effects).

Figure 6: This is interesting data, but also seems open to additional interpretations. While the introduction of the auditory cue could be perceived as action feedback, it may also act as a conditioned reinforcer and able to support actions independently. The authors raise the spectrum of cognitive as well as motor deficits in aging, but initiating actions under the control of conditioned reinforcement is a different hypothesis than self-initiated automatic action sequences. Additionally, the improved performance in aged mice with the addition of a loud salient auditory noise also raises the question of what action feedback is missing in Figure 1? Old mice in these age ranges, especially C57s loose large frequencies ranges in their hearing capacity. They appear to hear the white noise, but it does raise the potential that they are missing some other auditory cue that is there during sequence learning. Are they just hearing impaired and can't rely on any auditory feedback that normally arises during training (pellet dropping, lever, etc.)? This may not reflect anything on the ability to form sequences exactly, but to use auditory information?

Reviewer #3:

These data show action sequence chunking in young mice that is disrupted in aged mice. Aged mice show shorter action sequences that are, somewhat unexpectedly, sped up. This is associated with disrupted activation of caudolateral corticostriatal networks and is restored with the addition of a sequence termination feedback stimulus. Selective inactivation of DLS D1-expressing MSNs, like aging, also caused increased sequence speed. The behavioral analysis is novel and strong and is itself a significant contribution to the field. I particularly appreciate the inclusion of individual data. The ability to remedy the behavioral deficit with the feedback cue in aged mice is also a major strength of the report. The report is well-written and likely to make a significant contribution to several fields.

1) It is not clear how the DLS D1-MSN inactivation data relate to the aging data. Both findings are very interesting in their own right. But the manuscript (e.g., these statements from the Abstract: "Pharmacogenetic disruption of a striatal subcircuit in young mice reproduced most age-related action features" and Results "we hypothesized that age-related changes in corticostriatal function and resulting hypoactivity of dSPNs in the DLS could be generating deficits in action sequence latencies. We directly tested this possibility […]") implies that disrupted DLS D1 MSN activity underlies the deficits in aged mice. There are no data to support this and the DLS D1 MSN inactivation does not recapitulate the full or even most of the phenotype of aged mice. Gathering data to link these two data sets (e.g., evaluation of DLS D1 MSN activity in aged mice), as far as I can tell, is not feasible on any sort of reasonable timeframe. One option is to remove these data from the report. If they are included, then their presentation should be changed throughout to make clear this is not an evaluation of the mechanism of the aging effect and the limitations linking this result to the aging result should be thoroughly discussed.

2) Is it possible that differences in the ability to hear explain the differences in sequence performance in young vs. aged mice? The click of the pellet dispense provides a subtle cue signaling reward delivery. If the aged mice cannot hear this, they cannot use it as a cue to signal they should terminate pressing and check the magazine, as a result they have shorted sequences because they have to check the magazine more often. Data to support or refute this possibility are needed. Perhaps there are data from the literature to draw on. This possibility might also explain why the explicit auditory feedback cue that the aged mice can hear facilitates sequence duration in the aged mice.

3) Related to point 2, inspection of Figure 1 indicates that aged mice are checking the magazine more than the young mice. These data should be included and the possibility that the altered sequence performance is secondary to altered reward checking behavior should be discussed.

4) What is the rationale for analyzing only the 5 fastest sequences for the main data (Figure 3), but all sequences for the D1-MSN inactivation data (Figure I, J)? It would be preferable to analyze the sequence speed and IPI comparison for all sequences for both data sets.

5) Please include a quantification and statistical comparison of p-MAPK following CNO/GBR12783 on the hM4Di expressing v. non-expressing hemisphere, or inside v. outside the hM4di-mCherry expression zones.

6) Please include individual data as in Figure 1 for the DLS D1 MSN inactivation data presented in Figure 5.

---

## [Author Response]

Essential revisions:The reviewers identified 3 areas that required revisions that were deemed essential for publication. The first and most important is to clarify that the behavior shown by young rats is in fact chunked and that this is lost in aged rats. While the current detailed analysis was appreciated by all 3 reviewers, it lacks some of the traditional measures to show this. This is captured most clearly by the comment by reviewer 2 on sequence variability in Figure 1 and also analysis of Figure 3.

This is an important point and we agree that better measurements of chunking could have been provided in our initial version. We have now included new analyses of our data to adequately measure differences in the generation and development of action sequences in young and aged mice (see response to specific comments from reviewers below). Our new analysis of action chunking provides unprecedented detail to the study of sequence learning, and confirms and extends our previous finding that action sequence learning is dramatically impaired in aged mice.

The second critical area was to address whether the deficit shown is primarily in sequence generation or whether this is secondary to another problem, such as inability to hear the pellet dispenser or differences in the use of the trigger in the sequence in aged rats or perhaps simply locomotor problems.

We have now experimentally ruled out that aged mice are impaired in detecting pellet delivery. These results have been included in Figure 1—figure supplement 2. In addition, we did not observe overall decreased performance of aged animals as would be predicted by a conditioned reinforcer scenario (see response to reviewer 2 below). Thus, we believe that there should be no doubt that the deficit shown is primarily in sequence generation.

Finally there were concerns about the dose of CNO used and also generally the differences between the pattern of effect of the deficit in this experiment versus what is shown by the aged rats. These must all be addressed to satisfy the reviewers.

The reviewer’s concern on the dose of CNO is specifically addressed in our response to reviewer 2 below. We have now included a new analysis to address more directly which aspects of the aged behaviour are reproduced in the chemogenetics experiment, and which ones are not. These are detailed below. We did not intend to claim that this manipulation in young mice mimicked the aged brain. We have rephrased our text throughout to clarify that this experiment provided mechanistic substrate to only part of the problem in aged mice.

Reviewer #1:[…] One question I had is whether there is any relationship between the aged rats behavior and the unusual requirement for a short burst of presses to access the RR20 routine. I am not sure I understand fully what this means and its implications, but it seemed to me initially that the microchunks could be influenced by this unusual requirement. Is there any possibility of this?

It is important to clarify that we did not apply any temporal limits: the sequence trigger (ST) does not require to speed up lever press responses but only to execute them consecutively. In our RR20+ST task, the best behavioural strategy is to sustain actions for an extended period of time, irrespective of the speed of action – the generation of action sequences and related automaticity on the way occur spontaneously. Young mice comfortably incorporated the ST requirement, while aged mice could only manage to extend their actions for a few seconds, which allowed them to feed only on the low proportion of “low requirement” trials that were randomly picked by the RR schedule. But even in those, there was never a timeout to access the reward. Therefore, the temporal constraints and the speed established during microchunking are indeed evidenced by the ST (e.g. see new panel C on Figure 1), but cannot be a direct consequence of its unusual requirements. Rather, we interpret this microchunking as the inability of aged mice to temporally sustain actions. We have now introduced a diagram (new Figure 1—figure supplement 1) clarifying the ST procedure, and expanded the explanation of what it does in the Materials and methods (subsection “Standard instrumental training”).

Second I think the authors need to do a better job clarifying the relationship they see between the aging deficit and the effects of the chemogenetic manipulation. It reproduced some features of the aged mice, but it did not cause the loss of efficiency or (oddly) the pattern of IPI's between lever presses versus magazine-lever interactions (panels K). This seems like it deserves to be noted more clearly in the results (assuming I am not misunderstanding) and hit more head on in the discussion. Obviously there is no reason to require one specific pathway to account for all the changes in aging. Just saying this more clearly would be helpful, unless there is a more specific or different explanation.

With the introduction of quantitative chunking analyses in the study (see further comments below), we have now clearly identified which specific aspects of the aged behaviour are reproduced in the chemogenetics experiment, and which ones are not. Our new data revealed that interrupting intracellular signalling in dSPNs of the DLS during periods of sequence development can increase the speed of individual action sequences, although the number of elements fitted within each sequence as well as the extra sequence intervals remained unaltered. Therefore, this manipulation reproduced within-sequence but not extra-sequence features of action chunking in ageing, which, we argue, could explain the increased—rather than reduced—ability of dSPN-hM4Di^(+)^ mice to obtain rewards during ST training. This has now been clearly explained in our Results section and a paragraph has been included in the Discussion (paragraph six).

Lastly I think the authors should make clearer in the discussion why one would think the microchunking in the aged mice is a form of habit or model-free learning. They discuss nicely why their data suggests it is not. But there was not much of a lead in to this section to convince me this was an important question.

We based this discussion on the suggestion that limited computational resources can prioritise model-free over model-based operations. This is proposed in studies such as Otto et al. 2013, where they limited the working memory (WM) capacity of healthy humans through application of an acute stressor. They found that WM compromise in this way attenuated the contribution of model-based decisions to behaviour, but did not alter model-free processing. The reason why one would think the microchunking behaviour in the aged can encourage model-free behaviours in the first place is that it is well known that the aged brain has substantial working memory limitations (e.g. Bo et al., 2009; Wang et al., 2011; Wingfield et al., 1988). We argue, however, that different features in our data suggest that this is not the case. We have now expanded this discussion in the text to make this point clearer (Discussion, first paragraph).

Reviewer #2:[…] Overall, while the data approach was interesting, it largely appears that older mice might just be checking more, or at least aging predisposes an increase in checking behaviors.

We see the anticipatory response of checking the magazine as an unavoidable part of the instrumental program, and in fact we used these data to define the different elements of the action sequences throughout the study. While it is true that aged animals display, overall, more magazine checks than younger mice by the end of training, this is a reflection of their overt increase in the number of shorter action sequences executed throughout the session. In the new experiment presented in Figure 1—figure supplement 2, we submitted food-restricted animals to 6 days of magazine training, where no instrumental action was required to access to the food pellets (they were delivered randomly). During this time, animals learn that the discriminative stimulus (likely dispenser vibration) predicts the delivery of the outcome (food pellet), and accordingly produce more anticipatory responses, particularly immediately after the stimulus (see Figure 1—figure supplement 2). As shown in Figure 1—figure supplement 2, we recorded the same increase of anticipatory responding (magazine check rates) in both young and aged groups. From these data, we can assume that whatever differences of magazine check rates arise during instrumental training may be consequence of differences in the way instrumental programs are established, rather than different baselines of anticipatory behaviour.

[…] Figure 1: In relation to Figure 1, necessary data and analyses for interpretation are missing. Do young and old mice differ on the amount of sequences performed?

Yes, they do. The conclusion Figure 1 was intending to reach is that aged mice, despite showing the same overall LP performance (previous Figure 1, current Figure 1—figure supplement 1), organise their behaviour into a larger number of sequences that are much shorter in length (number of elements) and duration (seconds). This is very evident in our plots in Figure 1. In order to clarify that aged animals do produce a larger number of sequences, we have now substituted the panel in 1C by a graph showing average sequence rates (i.e. # sequences per min) developed throughout training. Corresponding results and statistics have been integrated in the new version of the manuscript, and we have highlighted this point in the Results section (paragraph one).

Data should be included on how well mice were able to meet the sequence trigger (the efficiency graph on Figure 3 addresses overall performance, but not performance of the ST). It is possible that the aged mice never learned the contingency between the ST and access to the RR, particularly as age is known to engender behavioral inflexibility, and pre-training without the ST requirement may have prevented its acquisition.

We believe that it is especially the aged group that undergoes strong learning of the ST, while young mice may passively overcome it (so learning in this latter group may be weak). We have now directly evidenced this by plotting the number of sequences that meet the sequence trigger (ST5) from day 10 (first day of ST5) onwards in both age groups (see new Figure 1—figure supplement 1 and related text). Data clearly show that only aged mice increased the number of sequences meeting the ST, whereas young animals kept those numbers stable until the end. This is supported by an overall significant effect of training (F_(3.065,42.908)_ = 7.757, p < 0.001) and a significant training x age interaction (F_(3.065,42.908)_ = 6.027, p < 0.01) (Supplementary file 1: StatsReport26). These results and corresponding statistical analysis have now been integrated in the manuscript.

It would similarly be of interest to examine sequence behavior in these two groups prior to the introduction of the ST requirement.

Behavioural structure data corresponding to day 7 (pre-sequence) has now been incorporated in the manuscript (see new Figure 1—figure supplement 1 and corresponding text).

In addition, the authors should include analysis and discussion of how variable their sequence behavior is. In Figure 1 it appears as though, even by day 17 of training, there is substantial variability associated with the sequences length and duration, both within and across subjects. To what extent are the mice actually performing stereotyped sequences vs. variable ones? This is an important concern, since automaticity would be expected to lead to relatively invariable sequences, yet it appears as though the sequences are highly variable, and therefore, it is unclear how mice are understanding and solving the task.

We disagree with the reviewer in that the establishment of automatic action sequences involves a homogenisation of the length of these sequences. A highly automatised behavioural chain (such as, for example, pedalling on a bicycle) can be sustained for as long as it is required by the current task. We purposely modelled this aspect of automatic behaviour by applying pseudorandom, rather than fixed, schedules of reinforcement. Therefore, the variability of sequence length observed in our data is likely to be the result of the variable requirement of the RR schedule, rather than within and between subjects variabilities. However, we do share with the reviewer that other reductions of variability throughout training can be used to quantitatively reflect automatisation.

Also missing are appropriate post hoc analyses that would arise from the significant interactions reported for D and E, which may affect the conclusions proposed. For example, a significant interaction is reported for Figure 1, and indeed the magnitude of sequence length is visually different between young and aged. However, all additional follow up analyses were not performed. They report a main factor of day for both young and old, but no comparison is made between young and aged for the different days. It seems likely that a difference would arise between ages on Day 17. This could then raise an additional hypothesis that aged mice do not extend sequences in the same manner, and importantly, and such, changes in the variability of such sequences needs to be examined.

We agree with the interpretation of the reviewer and believe is the same we were intending to provide. As we mention in our text, both young and aged mice were able to extend the number of execution elements in their sequences throughout training. This is supported by an overall effect of training (two-way mixed ANOVA F_(1,14)_ = 158.282, p < 0.001) (Figure 1). However, as indicated by a significant day x age interaction (F_(1,14)_ = 40.03, p < 0.001), this effect is of a different magnitude (i.e. aged animals extend *less* their sequences). We therefore ran subsequent simple effects analysis (one-way RM ANOVA in young and aged separately) to ensure that, despite this interaction, both groups were indeed extending the number of execution elements in each sequence. We believe that the significant interaction (rather than between subjects comparisons) is the most appropriate statistical measure to demonstrate that the increase of sequence length was significantly different across groups (see Supplementary file 1: StatsReport2 for full statistical analysis of these data). We have now rephrased our text (Results, paragraph one) to make sure that this interpretation is not missed.

Figure 3: The chunking analyses are quite interesting, however, they rely on visual interpretation of differences. The authors should apply some quantitative methods to these data to assess group differences. For instance, visually (figure supplement) it appears as though aged mice simply perform many more sequences than young mice (see concern above about amount of sequences), which might explain some of the chunking results. Proportional analyses may be appropriate here. As it stands, this data does not provide evidence that aged and young animals chunk sequences differently (and may suggest different amounts of overall lever pressing). Average inter-event intervals of various types could also be compared and would be informative.

We agree with the reviewer that more quantitative data should be used to evidence chunking. We have now substantially modified our dataset and provide straightforward measures of chunking throughout the manuscript (see new panels G-K in Figure 3 and its figure supplement 2; new Figure 5 and new Figure 6 and all related text). In order to better study age-related differences in chunking, we have categorised the action repertoire into two classes of elements: within-sequence elements (inter-press-intervals [IPIs] that reside within each sequence) and sequence boundary elements (a combination of press-check and check-press intervals that lay in-between sequences) (see expanded Materials and methods). We have also modified the visual representation of chunking: we now combine scatter plots with kernel density curves to represent the distribution of all within-sequence IPIs and all sequence boundary intervals that occur in one session (e.g. see new Figure 3). These graphs continue to provide a good visual representation of the emergence of chunking during training (e.g. Figure 3), and allow extracting different quantitative measures of chunking (new Figure 3).

With respect to the proportionality of analyses, we agree that in our initial representation aged mice had more data points than young mice as they took longer to finish the final sessions, and understand the reviewer’s concern for interpretation. In the new panels J and K in Figure 3, we have now plotted all the action element intervals (within-sequence and sequence boundary) performed by all young and all aged mice during the first 600 seconds of day 18 (uninterrupted sequence performance, see Materials and methods). These data are now proportional, as they include an equal period of continuous instrumental performance. Note how the overall number of within-sequence elements is similar in both age groups, whereas aged mice produce a higher number of boundary elements, which reflects a higher overall number of shorter sequences (see inset graphs in panels 3J and K).

Overall, our new analysis evidences clearly different automatisation processes shaping within-sequence and sequence boundary elements in young and aged mice. While young mice appear to exclusively chunk the elements comprised within the sequences, aged mice show evidence of early chunking of boundary elements. We refer the reviewer to the re-written paragraph in the Results section (subsection “Sequence learning leads to rapid response bursts and aberrant chunking in aged mice”) where these new results are laid out, which further support and extend our initial conclusion that aged mice develop aberrant automaticity characterised by ultrafast microchunks.

Figure 4: The investigation into how recruitment of activity in corticostriatal networks represented in Figure 4 is somewhat troubling, in that it may be inherently confounded by general differences in lever pressing and have nothing to do with sequence generation. Are the groups matched for total lever presses? Is it that one group (aged) is just making more lever presses and hence, that could account for the differential activation in the circuit (based on action execution and nothing to do with sequence generation)? In Figure 4, the large variability between subjects is apparent, and while various sequence factors were correlated with counts observed, whether overall lever presses correlated was not presented. Given this variability within a given group, do similar relationships arise for within group correlations?

Yes, our groups are matched in time of performance and overall amount of lever presses. On day 18, we submitted both young and aged groups to an extra session of training with uninterrupted instrumental performance for 20 minutes prior to transcardial perfusion. Aged mice did not produce more lever presses during this time (mean ± SEM, young: 949.62 ± 50.7; aged: 968.5 ± 82.8), as confirmed by the lack of statistical difference (t_(15)_ = -0.194, p = 0.849). We have now included in our correlation matrix (Figure 4) an extra row showing the correlation of the total amount of presses with the activation recorded in each region (see new panel D, “Total LP” at the bottom). None of these new correlations was significant (Figure 4 value inset), which supports that the significant correlations initially found in postero-lateral areas were related to sequence generation.

Further, for the analyses in D and E, how is the variability in each animal factor into the variability in the overall analyses? This is not clear in the Materials and methods. Are total activity levels normalized is some manner?

In our correlation analyses, the neuronal activation level in each of the regions was correlated with either sequence performance (Figure 4) or activation in every other region (Figure 4) of that same animal. Therefore, data points in each correlation of the matrix were obtained by contrasting within-individual measures, so normalisations across individuals are not suitable in this case. We have now clarified this in the figure legend and Materials and methods.

Figure 5: The CNO dosage used is quite high (3mg/kg), and this coupled with repeated dosing across days raises concerns about the potential changes in plasticity that may have occurred. This would raise alternative hypotheses other than an effect of altered D1 SPNs underlying sequence deficits in older animals.

We understand the reviewer’s concern on the dose of CNO, especially considering that it is a close molecular derivate of the highly psychoactive drug clozapine. We have however established in our laboratory that in order to achieve clear inhibitory effects on hM4Di-transduced neurons, the effective dose of systemic CNO needs to be pushed up to 3 mg/kg (see Figure 5—figure supplement 1). We are well aware of eventual reminiscent clozapine-like effects derived from dosing CNO up, and this is why we introduced in our study the littermate (rather than vehicle) controls, which received the same amount of CNO throughout training. Clozapine is a pro-cataleptic drug at high doses, which can have clear hipolocomotion effects through blockade of D2 receptors (see for example Valjent, Bertran-Gonzalez et al. *Neuropsychopharmacology*, 2011). Our data did not show any reduction of overall performance from day 9 (last day of no CNO) to day 10 (first day of CNO) of training (see Figure 5—figure supplement 1). Moreover, CNO treatment did not modify the slope of the instrumental acquisition curve in any of the groups, which was very similar to that obtained in drug-free mice (see Figure 1—figure supplement 1). Although this does not rule out that subthreshold clozapine activity may be taking place during this period, this undetectable side effects are also incorporated in our control mice, and therefore the differences detected between these two groups are likely driven by the chemogenetic manipulation.

The need for a relatively high dose of systemic CNO (3 mg/kg in our case) could be explained by the data recently published in Gomez & Bonaventura et al. *Science* 357, 503-507 (2017). In this work, the authors suggest that the main specific effects in DREADDs are driven by the small fraction of CNO that is peripherally converted to clozapine (subthreshold levels), provided that CNO itself is unable to penetrate the BBB. These authors show that the conversion of CNO to clozapine is rather inefficient, although a small concentration of peripheral clozapine appeared to be sufficient for substantial DREADD receptor occupancy. They also demonstrate that even very high doses of CNO (10 mg/kg) do not decrease baseline locomotion. Therefore, the conversion to clozapine in DREADDs seems to be the solution, rather than the problem, although this conversion appears too poor for clozapine-like side-effects to be detectable.

If specifically, D1 SPNs control over sequence execution is altered in aged mice, then acute attenuation should result in some alteration in performance. Also, it is unclear from the present data if dMSNs in the DLS are important in learning/acquiring these sequences, in their performance, or both.

It is important to clarify that we applied the CNO treatment to prevent the engagement of dorsolateral striatal plasticity at the time of sequence generation (i.e. from day 10 onwards), in an attempt to model the staticity of corticostriatal activation suggested in our experiments with aged mice. We therefore interpret these data as an enduring learning effect rather than an acute effect on performance. In other words, for action sequences to be appropriately established throughout training, it is required that plasticity in corticostriatal circuitry progressively expands to lateral territories, in accordance with previous reports (see refs Balleine 2005, Yin 2009 and Balleine & O’Doherty 2010 in our text). Under this reasoning, in our experiments where the circuitry has failed to mature in this way, sudden activation of dSPNs in the DLS of aged mice is unlikely to extend sequence duration or reduce speed, and, similarly, interruption of CNO treatment in dSPN-hM4Di^(+)^ mice is unlikely to reduce sequence duration or increase speed.

Although not unconvincing, for Figure 5, the authors should have consistency between the data presented here and in Figure 3 and C. In Figure 3, the authors show IPI and speed data from only the fastest sequences, while in Figure 5 the speed and IPI data appears to be collapsed across all sequences. Does CNO replicate the effect on the fastest sequences, and additionally, does age have the same overall effect as CNO appears to?

We have now homogenised the analyses of speed in the two experiments (see new Figure 3 and new Figure 5). In the ageing experiment, overall sequence speed is higher in the aged group, although the effect falls below threshold of significance (p = 0.054) (new Figure 3). However, frequency distribution of IPIs clearly showed a much larger number of very short IPIs in aged mice (new Figure 3), which prompted us to perform the fastest sequence study (Figure 3). In the DREADDs experiment, dSPN-hM4Di^(+)^ mice did show a similar overall increase in speed, although this time the effect was significant (Figure 5). The IPI distribution analysis also showed that dSPN-hM4Di^(+)^ mice produced more very short IPIs (new Figure 5), although this distribution was not as disproportionate as the one observed in aged mice, and therefore we did not consider relevant a fastest sequence study in this case. This is now clarified in the text: “This [increase in speed] was similar to the effect observed in aged mice (see Figure 3), although peak speeds within fast sequence bursts in dSPN-hM4Di^(+)^ were not as high as the ones recorded in aged mice”.

Is there any effect on the overall number of sequences?

Yes, these data are now included in the new Figure 5.

Why was the comparison made at day 10 instead of 7 (different schedule requirements were in place)?

As discussed above, the aim of the DREADDs experiment was to interrupt the engagement of lateral corticostriatal plasticity to attempt to recapitulate the unusual sequence features observed in aged mice, and so we compared the performance expressed on day 10 (expected start of sequence implementation and first day of CNO treatment) and day 17 (last day of CNO treatment). During this period, the ratio requirements were kept constant (RR20 + ST) to minimise interference with the CNO effect that was expected to develop across treatment.

Perhaps the authors should temper their comparisons between age and dMSN inhibition. Particularly since CNO did not recapitulate the effects of age on either action chunking or task performance (where indeed, CNO and age had opposite effects).

We did not intend to claim that chemogenetic manipulation of dSPNs mimicked all the deficits in the behaviour of aged mice but we did show that it reproduced some important aspects of action chunking.

Figure 6: This is interesting data, but also seems open to additional interpretations. While the introduction of the auditory cue could be perceived as action feedback, it may also act as a conditioned reinforcer and able to support actions independently. The authors raise the spectrum of cognitive as well as motor deficits in aging, but initiating actions under the control of conditioned reinforcement is a different hypothesis than self-initiated automatic action sequences.

The reviewer suggests an interesting interpretation of our data by which the stimulus (S), by being presented close to the delivery of the outcome (O), could act as a conditioned reinforcer driving the instrumental actions (i.e., A-S(O)). This possibility predicts, however, that, removing S should cause the instrumental actions to extinguish over time (at least partially). However, our data shows no evidence of a change in the rate of instrumental performance when S is removed (Figure 6, day 19 onwards). In fact, overall performance (LP per min) increased at the same rate as animals for whom S was retained (Figure 6). Therefore, our interpretation is that the presentation of S at the end of the sequences strengthens A and so the A-O contingency. The loss of S causes a temporal reorganisation of the instrumental chains, but does not precipitate their extinction.

Additionally, the improved performance in aged mice with the addition of a loud salient auditory noise also raises the question of what action feedback is missing in Figure 1? Old mice in these age ranges, especially C57s loose large frequencies ranges in their hearing capacity. They appear to hear the white noise, but it does raise the potential that they are missing some other auditory cue that is there during sequence learning. Are they just hearing impaired and can't rely on any auditory feedback that normally arises during training (pellet dropping, lever, etc.)? This may not reflect anything on the ability to form sequences exactly, but to use auditory information?

We report additional experiments to address this issue, which is related to the second concern point by reviewer 3. Please see answer below.

Reviewer #3:[…] 1) It is not clear how the DLS D1-MSN inactivation data relate to the aging data. Both findings are very interesting in their own right. But the manuscript (e.g., these statements from the Abstract: "Pharmacogenetic disruption of a striatal subcircuit in young mice reproduced most age-related action features" and Results "we hypothesized that age-related changes in corticostriatal function and resulting hypoactivity of dSPNs in the DLS could be generating deficits in action sequence latencies. We directly tested this possibility […]") implies that disrupted DLS D1 MSN activity underlies the deficits in aged mice. There are no data to support this and the DLS D1 MSN inactivation does not recapitulate the full or even most of the phenotype of aged mice. Gathering data to link these two data sets (e.g., evaluation of DLS D1 MSN activity in aged mice), as far as I can tell, is not feasible on any sort of reasonable timeframe. One option is to remove these data from the report. If they are included, then their presentation should be changed throughout to make clear this is not an evaluation of the mechanism of the aging effect and the limitations linking this result to the aging result should be thoroughly discussed.

We have now included new analyses that allow us to be more specific about which features of the aged behaviour are reproduced by dSPN manipulations in the DLS. We have also rephrased our text throughout to clarify that this experiment provided mechanistic substrate to only part of the problem in aged mice (see Abstract, Introduction, Results subheading and text, and Discussion). Although this experiment does not entirely reproduce the deficits observed in the aged, we believe that this manipulation provides support for the notion that dorsolateral striatum is involved in sequence learning.

2) Is it possible that differences in the ability to hear explain the differences in sequence performance in young v. aged mice? The click of the pellet dispense provides a subtle cue signaling reward delivery. If the aged mice cannot hear this, they cannot use it as a cue to signal they should terminate pressing and check the magazine, as a result they have shorted sequences because they have to check the magazine more often. Data to support or refute this possibility are needed. Perhaps there are data from the literature to draw on. This possibility might also explain why the explicit auditory feedback cue that the aged mice can hear facilitates sequence duration in the aged mice.

We believe that all animals are capable of detecting the delivery of the pellets, irrespective of their hearing capacity. We would like to clarify that, at least in our set up, there is not only an auditory sound (which is mostly masked by the fans providing continuous background noise), but also a very obvious vibration in the chamber due to the motor engagement in the pellet dispenser. We refer the reviewers to the new Figure 1—figure supplement 2, where we conducted a series of experiments specifically addressing this issue. In panel A, we recorded in parallel overall sound level (at different frequencies) and vibration inside the chamber while pellets were delivered at random times over a period of ~1.5 min. The data showed that although sound could predict the delivery of the pellets (at least at A frequency weighting), the vibration offered a much more salient signal. Considering that mice have at least two paws and tail on the grid floor at all times, and that they are exquisitely sensitive to mechanical stimulation through vibrissae detection of minute perturbations (e.g. Prescott et al. (2011) Scholarpedia, 6(11):6642), we are confident that pellet delivery is “sensed” rather than “heard” by the mice in the operant chambers. We directly tested pellet drop detection in a new experiment in young and aged mice by extracting the first check intervals (i.e. time elapsed between the delivery of the pellet and the subsequent magazine check) across 6 days of magazine training, where the pellets were randomly delivered into the magazine (Figure 1—figure supplement 2). Our data showed that both groups similarly reduced their first check intervals throughout training (panels B and D) while showing equivalent overall magazine check rates (panel C), thereby suggesting similar capacity to detect pellet deliveries. These results and corresponding data and statistics have been integrated in the new version of the manuscript.

3) Related to point 2, inspection of Figure 1 indicates that aged mice are checking the magazine more than the young mice. These data should be included and the possibility that the altered sequence performance is secondary to altered reward checking behavior should be discussed.

We see the anticipatory response of checking the magazine as an unavoidable part of the instrumental program, and in fact we used these data to define the different elements of the action sequences throughout the study. While it is true that aged animals display, overall, more magazine checks than younger mice by the end of training, this is a reflection of their overt increase in the number of shorter action sequences executed throughout the session. In the new experiment presented in Figure 1—figure supplement 2, we submitted food-restricted animals to 6 days of magazine training, where no instrumental action was required to access to the food pellets (they were delivered randomly). During this time, animals learn that the discriminative stimulus (likely dispenser vibration) predicts the delivery of the outcome (food pellet), and accordingly produce more anticipatory responses, particularly immediately after the stimulus (see Figure 1—figure supplement 2). As shown in Figure 1—figure supplement 2, we recorded the same increase of anticipatory responding (magazine check rates) in both young and aged groups. From these data, we can assume that whatever differences of magazine check rates arise during instrumental training may be consequence of differences in the way instrumental programs are established, rather than different baselines of anticipatory behaviour.

4) What is the rationale for analyzing only the 5 fastest sequences for the main data (Figure 3), but all sequences for the D1-MSN inactivation data (Figure I,J)? It would be preferable to analyze the sequence speed and IPI comparison for all sequences for both data sets.

Data analysis has now been homogenised across the two experiments.

5) Please include a quantification and statistical comparison of p-MAPK following CNO/GBR12783 on the hM4Di expressing v. non-expressing hemisphere, or inside v. outside the hM4di-mCherry expression zones.

We have now included this quantification and this dataset has been moved to the new Figure 5—figure supplement 1. Results (subsection “Chemogenetic inhibition of direct pathway projection neurons in dorsolateral striatum increases speed of action sequences”), source datasets and statistical analyses of this quantification have been integrated in the new version of the manuscript.

6) Please include individual data as in Figure 1 for the DLS D1 MSN inactivation data presented in Figure 5.

These data have now been included as Figure 5—figure supplement 2.